# Teleparallel Minkowski Spacetime with Perturbative Approach for Teleparallel Gravity on a Proper Frame

**Alexandre Landry** [1,*] and **Robert J. van den Hoogen** [2]

1    Department of Mathematics and Statistics, Dalhousie University, Halifax, NS B3H 4R2, Canada
2    Department of Mathematics and Statistics, St. Francis Xavier University, Antigonish, NS B2G 2W5, Canada; rvandenh@stfx.ca
*    Correspondence: a.landry@dal.ca; Tel.: +1-514-503-2051

**Abstract:** A complete perturbation theory suitable for teleparallel gravity is developed. The proposed perturbation scheme takes into account perturbations of the coframe, the metric, and the spin-connection, while ensuring that the resulting perturbed system continues to describe a teleparallel gravity situation. The resulting perturbation scheme can be transformed to one in which perturbations all take place within the co-frame. A covariant definition of a teleparallel Minkowski geometry is proposed. We compute the perturbed field equations for $f(T)$ teleparallel gravity and discuss the stability of the teleparallel Minkowski geometry within $f(T)$ teleparallel gravity.

**Keywords:** teleparallel gravity; teleparallel perturbation; constant torsion spacetimes; teleparallel Minkowski spacetime; perturbed teleparallel field equations; orthonormal perturbation; stability spacetime background

## 1. Introduction

There are two major classes of theories for physical phenomena: gravitational theories and quantized theories [1–4]. The first class of theories are used to explain phenomena at the astrophysical scale; for example, General Relativity (GR) has been very successful in explaining astrophysical phenomena [5–8]. However, the second class of theories concerns phenomena occurring at the microscopic scale involving fundamental quantum particles. Attempts have been made to reconcile the two classes of theories in order to have a general, all-encompassing theory. A theory that is capable of dealing with very low-amplitude physical and geometrical quantities, as is the case for theories based on quantization, is desirable.

Indeed, Quantum Mechanics (QM) as well as Quantum Field Theory (QFT) have well-established perturbative theories: a potential is perturbed, generating a correction of the eigenvalues of the energies, as well as corrections to the wave functions [1–4]. QM and QFT are well established and have been used to describe the gravitational corrections of curved spacetimes of physical phenomena that can occur at the microscopic scale [9–12]. Unfortunately, this perturbative approach to GR is problematic, primarily because one requires an identifiable background on which to perform the perturbations [13]. One can, of course, use gauge invariant variables to address this challenge.

Recently, there has been a growing interest in the development of teleparallel gravity as an alternative theory to GR [14–21]. Teleparallel gravity needs to be better understood and developed in order to address foundational, physical, and geometrical problems. Here, we will illuminate some of the challenges and nuances that are present within perturbative approaches to teleparallel gravity.

Golovnev and Guzman [22] studied a class of perturbations within a geometry having a Minkowski metric. They applied perturbations to a particular boosted coframe in which the metric has the Minkowski form and the torsion scalar is zero, but where the torsion tensor is non-zero. One may argue that any geometry in which the torsion tensor is non-zero

is inherently not a Minkowski geometry, but this is a matter of definition. In another paper, Jimenez et al. performed perturbations of Minkowski spacetime in $f(T)$ teleparallel gravity by using a trivial tetrad and having the perturbations encoded in infinitesimal Lorentz transformations [23]. Their approach, while correct, is restrictive when working towards a general perturbation theory within teleparallel gravity. In ref [24], the authors develop a complete perturbation theory that can be employed for perturbation analysis in Minkowski and flat Robertson-Walker-type cosmological spacetimes. Our analysis provides a different perspective and can be used as a general framework, and therefore, it complements the work in ref [24].

Recently, within a cosmological setting, Bahamonde et al. [25] investigated perturbations occurring on a FLRW-type background. They defined a very specific form for the perturbation compatible with this background. They then obtain the perturbed field equations. In addition, they investigated the consequent effects of perturbations on the torsion and on different physical quantities. Most of the types of perturbations studied lead to the flat FLRW background case under some precise limits. On the other hand, some perturbation modes do not propagate, which maintains the strong coupling. This is the case of the scalar and the pseudo-scalar parts of the perturbations. Here, we still have work with a limited scope; hence, the need for a more general theory of perturbations in teleparallel gravity.

Bamba and Cai's papers focus on Gravitational Waves (GWs) in teleparallel gravity [26,27]. GWs are a class of wave-like perturbations of Minkowski spacetime. They are still dealing here with a specific case of perturbation. In Bamba [26], they place themselves in the Minkowski background to process the GWs in teleparallel gravity. In Cai [27], they place themselves in the FLRW background. They therefore have a generalization of Bamba's work for GWs that are compatible with the cosmological models. In addition, in [27], they add the effects of scalar fields in their perturbations. Not only are they still dealing with specific cases of perturbations, but they are moving from the Minkowski background to the FLRW background. However, they still do not have a general theory for the Minkowski background. Therefore, a more general and fundamental theory that is applicable for any perturbation and any co-frame in Minkowski spacetime in teleparallel gravity is needed.

We begin this paper with a definition of Minkowski geometry and Minkowski spacetime within teleparallel gravity. Then, we will investigate the effects of perturbations in teleparallel gravity. After, we will study the stability of Minkowski spacetime by using the perturbed quantities and field equations.

In teleparallel gravity, co-frames encode both the gravitational and inertial effects. Our goal is to explore the perturbations of gravity, and therefore, we shall carefully construct a perturbative theory that achieves this goal. If we transform initially to "proper" frames which encode only the gravitational effects and then perform perturbations on all physical quantities, consequently ensuring that the resulting perturbed theory is still within the class of teleparallel theories of gravity will yield the general allowable form for perturbations within teleparallel gravity. We will perturb the physical quantities which maintain the "proper frames", thus avoiding the challenge of interpreting the spurious inertial effects that may appear in "non-proper frames" [14–16,28,29].

We want to highlight the effects of perturbations in teleparallel gravity. For example, in an absolute vacuum, one can highlight the effects of perturbations modifying this same vacuum. For example, we will determine the gravitational Energy-Momentum associated with a perturbation. We will apply this theory of perturbations in teleparallel gravity to some examples and problems of Physics [16,30,31]. Particularly, we will study through these coframe perturbations the stability of the Minkowski background, and determine the required symmetry conditions to satisfy.

This paper is divided as follows. In Section 2, we present a summary of teleparallel gravity and propose a definition of Minkowski geometry within teleparallel gravity. In Section 4, we will define the perturbations maintaining the "proper frames", the orthonormal

framework, and we will also provide the perturbed Field Equations (FEs). In Section 5, we will explore some coframe perturbations to determine the stability criterions for Minkowski spacetime. We can also generalize these criterions to null and constant torsion spacetimes.

## 2. Teleparallel Theories of Gravity

### 2.1. Notation

Greek indices $(\mu, \nu, \dots)$ are employed to represent the spacetime coordinate indices, while Latin indices $(a, b, \dots)$, are employed to represent frame or tangent-space indices. As is standard notation, round parentheses surrounding indices represent symmetrization, while square brackets represent anti-symmetrization. Any quantity that is computed using a Levi-Civita connection $\overset{\circ}{\omega}{}^a{}_{b\mu}$ will have a circle above the symbol. A comma will denote a partial derivative. The metric signature is assumed to be $(-,+,+,+)$.

### 2.2. Torsion-Based Theories

Torsion-based theories of gravity are a subclass of Einstein-Cartan theories [15,16,32]. This superclass of theories contains theories based solely on the curvature, for example, General Relativity, or $f(R)$ theories where $R$ is Ricci curvature scalar. Einstein-Cartan theories of gravity also contain theories of gravity that are based solely on the torsion, for example, teleparallel theories of gravity, including New General Relativity [33] and $f(T)$ theories where $T$ is the torsion scalar. In addition, theories of gravity based on both the curvature and torsion scalars ($f(R, T)$-type) are also subclasses of the Einstein-Cartan theories of gravity. Recently, there has been an emergence of theories based on non-metricity ($f(Q)$-type), although they are less well known [16,34,35]. In this paper, we are interested in teleparallel gravity, and in particular, $f(T)$ teleparallel gravity [14–20,29].

### 2.3. Geometrical Framework for Teleparallel Gravity

Let $M$ be a 4-dimensional differentiable manifold with coordinates $x^\mu$. Then, the geometry of the manifold is characterized by the three geometrical objects.

- **The Co-frame:** $h^a = h^a{}_\mu dx^\mu$. This quantity generally encodes both the gravitational and inertial effects in a gravitational system. The dual of the co-frame is defined as the vector field $h_a = h_a{}^\mu \frac{\partial}{\partial x^\mu}$, such that $h^a{}_\mu h_b{}^\mu = \delta^a_b$.
- **The Gauge Metric:** $g_{ab}$. This object expresses the "metric" of the tangent space, such that $g_{ab} = g(h_a, h_b)$. Having a metric allows one to define the lengths and angles.
- **The Spin-connection:** $\omega^a{}_b = \omega^a{}_{b\mu} dx^\mu$. Having a connection allows one to "parallel transport'," or equivalently, it allows one to define a covariant differentiation.

In teleparallel gravity, the co-frame, gauge metric, and spin connection are restricted and interdependent, characterized by the following two postulates [14–16]:

- **Null Curvature:**

$$R^a{}_{b\nu\mu} \equiv \omega^a{}_{b\mu,\nu} - \omega^a{}_{b\nu,\mu} + \omega^a{}_{c\nu}\omega^c{}_{b\mu} - \omega^a{}_{c\mu}\omega^c{}_{b\nu} = 0 \tag{1}$$

- **Null Non-Metricity:**

$$Q_{ab\mu} \equiv -g_{ab,\mu} + \omega^c{}_{a\mu}g_{cb} + \omega^c{}_{b\mu}g_{ac} = 0 \tag{2}$$

In teleparallel gravity, the only remaining non-null field strength is the torsion defined as

$$T^a{}_{\mu\nu} = h^a{}_{\nu,\mu} - h^a{}_{\mu,\nu} + \omega^a{}_{b\mu}h^b{}_\nu - \omega^a{}_{b\nu}h^b{}_\mu \tag{3}$$

It is now possible to construct a gravitational theory that depends only on the torsion. However, before proceeding, we illustrate the effects of gauge transformations on the geometry, and how we can judiciously choose a gauge to simplify our computations.

### 2.4. Linear Transformations and Gauge Choices

From the Principle of Relativity, we impose the requirement that the physical gravitational system under consideration be invariant under $GL(4, \mathbb{R})$ local linear transformations of the frame. These types of transformations allow one to pass from one frame of reference to another frame of reference. For the fundamental geometrical quantities $\{h^a, g_{ab}, \omega^a{}_{bc}\}$, we have the following transformation rules under a general linear transformation $M^a{}_b \in GL(4, \mathbb{R})$:

$$h'^a{}_\mu = M^a{}_b h^b{}_\mu, \tag{4}$$

$$g'_{ab} = M_a{}^e M_b{}^f g_{ef}, \tag{5}$$

$$\omega'^a{}_{b\mu} = M^a{}_e \omega^e{}_{f\mu} M_b{}^f + M^a{}_e \partial_\mu M_b{}^e. \tag{6}$$

where $M_b{}^a = (M^{-1})^a{}_b$ represents the inverse matrix. Equation (6) shows that the Spin-connection transforms non-homogeneously under a general linear transformation.

#### 2.4.1. Gauge Choices and Teleparallel Gravity

Physical phenomena must respect the principle of Gauge Invariance. The physical phenomenon must be explainable and valid, regardless of the gauge and its possible transformations. If this general principle is important for quantized theories, then this same principle is also important for teleparallel gravity. Generally, we have a tremendous choice of gauge, depending on the assumed symmetries of the physical system. However, once we have made a gauge choice, the consequent field equations describing the theory must transform covariantly (i.e., they are invariant) under any remaining gauge freedom.

#### Proper Orthonormal Frame

The Null Curvature postulate guarantees that there exists an element $M^a{}_b \in GL(4, \mathbb{R})$, such that

$$\omega^a{}_{b\mu} \equiv (M^{-1})^a{}_b \partial_\mu (M^b{}_c) \tag{7}$$

Since the connection transforms non-homogeneously under local linear transformations, we can always apply the linear transformation $M^a{}_b$ to transform to a proper frame in which $\omega^a{}_{b\mu} = 0$. Further, within this proper frame, given the Null Non-Metricity postulate, it is then possible to apply a second constant linear transformation to bring the gauge metric to some desired form. For example, we can transform to a gauge in which the spin connection is null and the gauge metric is $g_{ab} = \text{Diag}[-1, 1, 1, 1]$, which we will call a "proper orthonormal frame". The only remaining gauge freedom in this case are global (constant) Lorentz transformations.

#### Orthonormal Frame

If one prefers not to be restricted to a proper frame, then there is more flexibility. Since the gauge metric is symmetric, we can still always choose an "orthonormal frame" in which the gauge metric becomes $g_{ab} = \text{Diag}[-1, 1, 1, 1]$, but where the spin connection may be non-trivial. Assuming an orthonormal frame, the remaining gauge freedom is represented by proper orthochronous Lorentz transformations in the $SO^+(1, 3)$ subgroup of $GL(4, \mathbb{R})$. Other gauge choices might include Complex-Null, Half-Null, Angular-Null, and others [17–19]. In the orthonormal frame, given the Null Curvature postulate, there exists a $\Lambda^a{}_b \in SO^+(1, 3)$, such that the spin connection is [36,37]:

$$\omega^a{}_{b\mu} \equiv (\Lambda^{-1})^a{}_b \partial_\mu (\Lambda^b{}_c) \tag{8}$$

and given the Null Non-Metricity postulate, we have the restriction $\omega_{(ab)\mu} = 0$.

However, in either choice of gauge, we note that the spin connection, $\omega^a{}_{b\mu}$, is not a true dynamical variable and that it only encodes inertial effects present in the choice of frame [14–20,28,29].

### 2.5. Action for $f(T)$ Teleparallel Gravity

In principle, one can construct a Lagrangian density from any of the scalars built from the torsion tensor. One such scalar is [14–20,29]:

$$T = \frac{1}{4} T^a{}_{bc} T_a{}^{bc} + \frac{1}{2} T^a{}_{bc} T^{cb}{}_a - T^a{}_{ca} T^{bc}{}_b,\tag{9}$$

which we will call "the" torsion scalar $T$. Another related scalar, for example, used in New General Relativity [33], is

$$\widetilde{T} = c_1 T^a{}_{bc} T_a{}^{bc} + c_2 T^a{}_{bc} T^{cb}{}_a + c_3 T^a{}_{ca} T^{bc}{}_b\tag{10}$$

Other torsion scalars could be included, but these scalars are not invariant under $SO^+(1,3)$, and they include parity violating terms [33].

Here, we are interested in a particular class of teleparallel gravity theories, $f(T)$ teleparallel gravity. The action describing the $f(T)$ teleparallel theory of gravity containing matter is [14–20,29]:

$$S_{f(T)} = \int d^4 x \left[ \frac{h}{2\kappa} f(T) + \mathcal{L}_{Matter} \right].\tag{11}$$

where $h = \mathrm{Det}\left(h^a{}_\mu\right)$ is the determinant of the veilbein, the parameter $\kappa$ is the gravitational coupling constant which contains the physical constants, and $f(T)$ is an arbitrary function of the torsion scalar $T$, given by Equation (9).

### 2.6. Field Equations for $f(T)$ Teleparallel Gravity

From the action integral expressed by Equation (11), we determine the field equations by varying with respect to the coframe $h^a{}_\mu$ [14–20,29]:

$$\kappa \Theta_a{}^\mu = \frac{f_T(T)}{h} \partial_\nu \left( h S_a{}^{\mu\nu} \right) + f_{TT}(T) S_a{}^{\mu\nu} \partial_\nu T + \frac{f(T)}{2} h_a{}^\mu - f_T(T) \left( \omega^b{}_{a\nu} + T^b{}_{a\nu} \right) S_b{}^{\mu\nu}.\tag{12}$$

The superpotential is defined as [14,15,17,18]:

$$S_a{}^{\mu\nu} = \frac{1}{2} \left( T_a{}^{\mu\nu} + T^{\nu\mu}{}_a - T^{\mu\nu}{}_a \right) - h_a{}^\nu T^{\rho\mu}{}_\rho + h_a{}^\mu T^{\rho\nu}{}_\rho.\tag{13}$$

The canonical Energy-Momentum is defined as [18]:

$$h \Theta_a{}^\mu \equiv \frac{\delta \mathcal{L}_{Matter}}{\delta h^a{}_\mu}.\tag{14}$$

Now, expressing the field Equation (12) in terms of the tangent-space components allows one to split the field equations into symmetric and antisymmetric parts. The symmetric and antisymmetric parts of the $f(T)$ teleparallel gravity FEs are respectively [17–19]:

$$\begin{aligned} \kappa \Theta_{(ab)} &= f_{TT}(T) S_{(ab)}{}^\mu \partial_\mu T + f_T(T) \overset{\circ}{G}_{ab} + \frac{g_{ab}}{2} [f(T) - T f_T(T)], \\ 0 &= f_{TT}(T) S_{[ab]}{}^\mu \partial_\mu T, \end{aligned}\tag{15}$$

where $\overset{\circ}{G}_{ab}$ is the Einstein tensor computed from the Levi-Civita connection of the metric.

We note with an orthonormal gauge choice, and consequent invariance under $SO^+(1,3)$ transformations, it can be shown that

$$\Theta_{[ab]} = 0, \tag{16}$$

and that the metrical energy-momentum $T_{ab}$ and the symmetric part of the canonical energy-momentum satisfy

$$\Theta_{(ab)} = T_{ab} \equiv \frac{1}{2} \frac{\delta L_{Matt}}{\delta g_{ab}}. \tag{17}$$

## 3. Constant Torsion Spacetimes

A class of interesting spacetimes are those leading to a constant torsion scalar, i.e., $T = T_0 =$ Const. This class of spacetimes includes the Minkowski spacetime, amongst others. In this case, the Equation (15) will simplify with $\partial_\mu T = 0$ as follows, leaving only the symmetric part of the field equations:

$$\kappa \Theta_{(ab)} = f_T(T_0) \, \overset{\circ}{G}_{ab} + \frac{g_{ab}}{2} \left[ f(T_0) - T_0 \, f_T(T_0) \right]. \tag{18}$$

The antisymmetric part of the field equations becomes identically satisfied. We can now divide Equation (18) by $f_T(T_0)$ to obtain:

$$\begin{aligned}
\kappa_{eff} \Theta_{(ab)} &= \overset{\circ}{G}_{ab} + g_{ab} \left[ \frac{f(T_0)}{2 \, f_T(T_0)} - \frac{T_0}{2} \right] \\
&= \overset{\circ}{G}_{ab} + g_{ab} \, \Lambda(T_0). 
\end{aligned} \tag{19}$$

where we define the re-scaled gravitational coupling constant $\kappa_{eff} = \frac{\kappa}{f_T(T_0)}$ and an effective cosmological constant $\Lambda(T_0)$, both dependent on the value of $T = T_0$. We observe that if $T = T_0 =$ Const, then the $f(T)$ teleparallel field equations reduce to those of GR, having a re-scaled gravitational coupling and a cosmological constant.

Due to its importance in characterizing the Minkowski geometry, we carefully consider the case of $T_0 = 0$ for further consideration.

### 3.1. Null Torsion Scalar Spacetimes

When $T_0 = 0$, the field equations reduce to:

$$\begin{aligned}
\kappa_{eff} \Theta_{(ab)} &= \overset{\circ}{G}_{ab} + g_{ab} \left[ \frac{f(0)}{2 \, f_T(0)} \right], \\
&= \overset{\circ}{G}_{ab} + g_{ab} \, \Lambda(0). 
\end{aligned} \tag{20}$$

where $\kappa_{eff} = \frac{\kappa}{f_T(0)}$ and $\Lambda(0) = \frac{f(0)}{2 f_T(0)}$. If $f(0) \neq 0$, then the Cosmological Constant $\Lambda(0) \neq 0$.

#### 3.1.1. Definition: Minkowski Geometry and Minkowski Spacetime

Before obtaining the field equations and introducing the perturbations on such, one must clearly define the true nature of the Minkowski spacetime in teleparallel gravity in a covariant way. This will make it possible to better understand the nature and origin of the equations involving the dominant quantities with respect to the perturbed quantities. This geometry is characterized as follows:

- Maximally symmetric: The Minkowski geometry is invariant under a $G_{10}$ group of transformations [18].
- Null Curvature: $R^a{}_{b\mu\nu} = 0$
- Null Torsion: $T^a{}_{\mu\nu} = 0$
- Null Non-Metricity: $Q_{ab\mu} = 0$

One of the consequences is that Minkowski geometry is everywhere a smooth geometry without singularity. This covariant definition of teleparallel Minkowski geometry has been proposed also by Beltran et al. [38].

We distinguish between Minkowski geometry and Minkowski spacetime in teleparallel gravity as follows. Minkowski geometry is defined independently of any field equations, while Minkowski spacetime is a Minkowski geometry that is a solution to the teleparallel gravity field equations where the matter source is a vacuum, $\Theta_{ab} = 0$.

If the geometry is Minkowski, then the torsion scalar is identically zero. Note that the converse is not necessarily true. The Einstein tensor $\overset{\circ}{G}_{ab} = 0$, and since the matter source is a vacuum, $\Theta_{ab} = 0$, the field Equation (20) reduce to

$$0 = \frac{f(0)}{2} g_{ab}. \tag{21}$$

From the field equation (21), if the geometry is Minkowski and $\Theta_{ab} = 0$, then $f(0) = 0$. In this case, the solution is a Minkowski spacetime, a Minkowski geometry that satisfies the field equations in vacuum. Alternatively, if $f(0) \neq 0$, then a solution to the field Equation (21) necessarily requires a non-null $\Theta_{ab}$, and consequently, this spacetime is not a Minkowski spacetime, even though the geometry is Minkowski. Of course, the non-trivial $\Theta_{ab}$ can be interpreted as the energy density of the vacuum. Expressing the statement clearly, Minkowski geometry is a solution to the vacuum $f(T)$ teleparallel gravity field equations only if $f(0) = 0$.

## 4. Perturbations in Teleparallel Geometries

### 4.1. Proper Orthonormal Perturbation of the Co-Frame

As described earlier, a teleparallel geometry is characterized in general via the triplet of quantities, the co-frame one form $h^a$, the spin connection one-form $\omega^a{}_b$, and the metric tensor field $g_{ab}$, with two constraints, Null Curvature and Null Non-Metricity. As argued earlier, assuming that the physical system is invariant under the $GL(4, \mathbb{R})$ linear transformations (see also ref. [38]), this means that even before constructing a perturbative theory, one can always choose to begin in a "proper orthonormal frame" as our background without a loss of generality:

$$h^a = h^a{}_\mu dx^\mu, \qquad \omega^a{}_b = 0, \qquad g_{ab} = \eta_{ab} = \text{Diag}[-1, 1, 1, 1]. \tag{22}$$

Now, we apply a perturbation to all three quantities, as follows:

$$h'^a = h^a + \delta h^a, \qquad \omega'^a{}_b = \delta\omega^a{}_b, \qquad g'_{ab} = \eta_{ab} + \delta g_{ab} \tag{23}$$

The perturbed geometry is no longer expressed in a proper orthonormal frame. The perturbed system is only proper if $\delta\omega^a{}_b = 0$, and orthonormal if $\delta g_{ab} = 0$. However, we shall show that we can always transform to a proper orthonormal perturbation scheme.

We note that the perturbed geometry given by the triplet $\{h'^a, \omega'^a{}_b, g'_{ab}\}$ must still satisfy the Null Curvature and Null Non-Metricity constraints or else one is moving outside of the theory of teleparallel gravity. In general, the perturbations $\delta h^a$, $\delta\omega^a{}_b$, and $\delta g_{ab}$ are not all independent. The Null Curvature constraint for the perturbed connection $\omega'^a{}_b$ implies that there exists some local linear transformation $L^a{}_b \in GL(4, \mathbb{R})$, such that

$$\delta\omega^a{}_b = (L^{-1})^a{}_c dL^c{}_b \tag{24}$$

where $d$ indicates the exterior derivative. This means that we can apply this general linear transformation to the perturbed system to express it in a perturbed proper frame

$$\bar{h}'^a = L^a{}_b(h^b + \delta h^b), \qquad \bar{\omega}'^a{}_b = 0, \qquad \bar{g}'_{ab} = (L^{-1})^c{}_a(L^{-1})^d{}_b(\eta_{cd} + \delta g_{cd}) \tag{25}$$

where we have used a bar to indicate that we are now in a proper frame.

The Null Non-Metricity condition applied to this "perturbed proper frame" (25) means that $\bar{g}'_{ab}$ is a symmetric matrix of the constants which can diagonalized. That is, there exists a matrix $P^a_b \in GL(4, \mathbb{R})$ of constants such that $\bar{g}'_{ab} = (P^{-1})^c_a (P^{-1})^d_b \eta_{cd}$. So, we can apply this constant transformation $P^a_b$ to the "perturbed proper frame" (25) to obtain a "perturbed proper orthonormal frame" without a loss of generality.

$$\hat{h}'^a = P^a_b \bar{h}'^b = P^a_b L^b_c (h^c + \delta h^c), \tag{26a}$$
$$\hat{\omega}'^a_b = 0, \tag{26b}$$
$$\hat{g}'_{ab} = \eta_{ab}. \tag{26c}$$

We observe that we can investigate perturbations in teleparallel geometries by simply looking at the perturbations in a co-frame, using proper orthonormal frames. Doing so ensures that the Null Curvature and Null Non-Metricity constraints are respected. If we define the compositions of the two linear transformations as matrix $M^a_b = P^a_c L^c_b \in GL(4, \mathbb{R})$, then the "perturbed proper orthonormal frame" becomes

$$\hat{h}'^a = M^a_b \left( h^b + \delta h^b \right). \tag{27}$$

which encodes all possible perturbations within a proper orthonormal framework. If $M^a_b = \delta^a_b$, then the only perturbations are perturbations in the original proper orthonormal frame. The matrix $M^a_b$ encodes the perturbations that took place originally in the spin connection and metric, but it ensures that the resulting perturbed system is teleparallel in nature. For completeness, the original perturbations can be expressed in terms of $M^a_b$, as

$$\delta \omega^a_b = (M^{-1})^a_c dM^c_b, \qquad \delta g_{ab} = (M^{-1})^c_a (M^{-1})^d_b \eta_{cd} - \eta_{ab} \tag{28}$$

Now, in a perturbative approach, to the first order, we have that

$$M^a_b \approx \delta^a_b + \mu^a_b \tag{29}$$
$$\delta h^a \approx \nu^a_b h^b \tag{30}$$

for some $\mu^a_b$ and $\nu^a_b \in \mathfrak{gl}(4, \mathbb{R})$. Therefore, putting it all together, we have to first order

$$\hat{h}'^a = h^a + (\mu^a_b + \nu^a_b)h^b = h^a + \lambda^a_b h^b, \tag{31a}$$
$$\hat{\omega}'^a_b = 0, \tag{31b}$$
$$\hat{g}'_{ab} = \eta_{ab}, \tag{31c}$$

where $\lambda^a_b \in M(4, \mathbb{R})$, the set of $4 \times 4$ real-valued matrices. Perturbations of the independent quantities in teleparallel geometry can always be transformed to the form (31). The matrix $\lambda$ can be invariantly decomposed into trace, symmetric trace-free, and anti-symmetric parts.

For the next section and in the appendix, we will apply the perturbations

$$\delta h^a = \lambda^a_b h^b, \qquad \delta \omega^a_b = 0, \qquad \delta g_{ab} = 0, \tag{32}$$

to the $f(T)$ teleparallel field equations in a proper orthonormal frame. In particular, we will look at perturbations of constant scalar torsion spacetimes.

*4.2. Perturbed f(T) Teleparallel Field Equations: General*

Considering the perturbations of the field Equation (15), we obtain

$$
\kappa \left[ \Theta_{(ab)} + \delta\Theta_{(ab)} \right] = f_{TT}(T + \delta T) \left[ S_{(ab)}{}^{\mu} + \delta S_{(ab)}{}^{\mu} \right] \left[ \partial_\mu T + \partial_\mu(\delta T) \right]
$$
$$
+ f_T(T + \delta T) \left[ \overset{\circ}{G}_{ab} + \delta\overset{\circ}{G}_{ab} \right]
$$
$$
+ \frac{g_{ab}}{2} \left[ f(T + \delta T) - (T + \delta T) f_T(T + \delta T) \right], \tag{33a}
$$
$$
0 = f_{TT}(T + \delta T) \left[ S_{[ab]}{}^{\mu} + \delta S_{[ab]}{}^{\mu} \right] \partial_\mu(T + \delta T), \tag{33b}
$$

which to the first order in the perturbations yields

$$
\kappa\,\delta\Theta_{(ab)} \approx \left[ f_{TTT} S_{(ab)}{}^{\mu} \partial_\mu T + f_{TT} \left( \overset{\circ}{G}_{ab} - \frac{T}{2} g_{ab} \right) \right] \delta T + f_T\,\delta\overset{\circ}{G}_{ab}
$$
$$
+ f_{TT} \left[ \delta S_{(ab)}{}^{\mu} \partial_\mu T + S_{(ab)}{}^{\mu} \partial_\mu(\delta T) \right] + O\left(|\delta h|^2\right), \tag{34a}
$$
$$
0 \approx f_{TTT} \left[ S_{[ab]}{}^{\mu} \partial_\mu T \right] \delta T + f_{TT} \left[ S_{[ab]}{}^{\mu} \partial_\mu(\delta T) + \delta S_{[ab]}{}^{\mu} \partial_\mu T \right] + O\left(|\delta h|^2\right), \tag{34b}
$$

where we no longer explicitly show functional dependence in *F*.

In Appendix A, perturbations of different dependent quantities are explicitly computed in terms of the perturbations (32), for example $\delta T, \delta S_{[ab]}{}^{\mu}$, etc. Here, $\delta T$ is given by Equation (A8) and $\delta S_{ab}{}^{\mu}$ is given by Equation (A13). Equation (34) gives us expressions for the perturbations to the matter resulting from the perturbations in the co-frame, and constraints on the perturbations to the antisymmetric part of the super-potential.

*4.3. Perturbed f(T) Teleparallel Field Equations: Constant Torsion Scalar*

To study the effects of perturbations of the co-frame in constant torsion scalar spacetimes, one substitutes $T = T_0 = $ Const into Equation (34). This means $\partial_\nu T = 0$. If we divide by $f_T(T_0)$, Equation (34) becomes:

$$
\kappa_{eff}\,\delta\Theta_{(ab)} \approx \delta\overset{\circ}{G}_{ab} + \frac{f_{TT}(T_0)}{f_T(T_0)} \left[ S_{(ab)}{}^{\mu} \partial_\mu(\delta T) + \delta T \left( \overset{\circ}{G}_{ab} - \frac{T_0}{2} g_{ab} \right) \right] + O\left(|\delta h|^2\right), \tag{35a}
$$
$$
0 \approx \left( \frac{f_{TT}(T_0)}{f_T(T_0)} \right) S_{[ab]}{}^{\mu} \partial_\mu(\delta T) + O\left(|\delta h|^2\right), \tag{35b}
$$

where $\kappa_{eff} = \frac{\kappa}{f_T(T_0)}$. In general, $S_{[ab]}{}^{\mu} \neq 0$, and therefore, the perturbations in the torsion scalar are constant. Of course, in situations in which some component of $S_{[ab]}{}^{\mu} = 0$, and then the corresponding $\partial_\mu(\delta T) \neq 0$.

*4.4. Perturbed f(T) Teleparallel Field Equations: Zero Torsion Scalar*

For spacetimes that have a zero torsion scalar, $T = 0$, and Equations (35a) and (35b) become:

$$
\kappa_{eff}\,\delta\Theta_{(ab)} \approx \delta\overset{\circ}{G}_{ab} + \frac{f_{TT}(0)}{f_T(0)} \left[ S_{(ab)}{}^{\mu} \partial_\mu(\delta T) + \delta T\,\overset{\circ}{G}_{ab} \right] + O\left(|\delta h|^2\right), \tag{36a}
$$
$$
0 \approx \left( \frac{f_{TT}(0)}{f_T(0)} \right) S_{[ab]}{}^{\mu} \partial_\mu(\delta T) + O\left(|\delta h|^2\right), \tag{36b}
$$

where $\kappa_{eff} = \frac{\kappa}{f_T(0)}$. As before, in general, $S_{ab}{}^{\mu} \neq 0$, and therefore, the perturbations in the torsion scalar are constant. These equations represent perturbations in non-Minkowski but zero torsion scalar spacetimes. However, they can reduce to perturbations of the $f(T)$ teleparallel field equations with a teleparallel Minkowksi geometry when $S_{ab}{}^{\mu} = 0$

and $\overset{\circ}{G}_{ab} = 0$, which are the conditions that are compatible with a teleparallel Minkowski spacetime, as defined in Section 3.1.1.

### 4.5. Perturbed $f(T)$ Teleparallel Field Equations: The Zero Torsion Scalar Perturbation Limit

We are curious to know what happens in the restricted perturbation scheme in which $\delta T \rightarrow 0$ only. Starting with Equation (34), we take the limit $\delta T \rightarrow 0$, and these perturbed field equations become:

$$\kappa\,\delta\Theta_{(ab)} \approx f_T\,\delta\overset{\circ}{G}_{ab} + f_{TT}\Big[\delta S_{(ab)}{}^{\mu}\,\partial_{\mu}T + S_{(ab)}{}^{\mu}\,\partial_{\mu}(\delta T)\Big] + O\Big(|\delta h|^2\Big), \tag{37a}$$

$$0 \approx f_{TT}\Big[\delta S_{[ab]}{}^{\mu}\partial_{\mu}T + S_{[ab]}{}^{\mu}\partial_{\mu}(\delta T)\Big] + O\Big(|\delta h|^2\Big). \tag{37b}$$

Looking at Equation (37b), given that in general, $S_{[ab]}{}^{\mu} \neq 0$ and $\delta S_{[ab]}{}^{\mu} \neq 0$ (or equivalently, the torsion tensor and perturbations of the torsion tensor are non-trivial, respectively), we observe that if the torsion scalar is not constant, $\partial_{\mu}T \neq 0$, and then the perturbations of the torsion scalar are also not constant, that is, $\partial_{\mu}(\delta T) \neq 0$. Conversely, if $\partial_{\mu}T = 0$, then $\partial_{\mu}(\delta T) = 0$.

### 4.6. Perturbed $f(T)$ Teleparallel Field Equations: Minkowski

For the Minkowski spacetimes, as defined in Section 3.1.1, since the torsion tensor is zero by definition, the superpotential terms $S_{(ab)}{}^{\mu} = S_{[ab]}{}^{\mu} = 0$. Further, the Einstein tensor $\overset{\circ}{G}_{ab} = 0$, and as argued before, $f(0) = 0$, so that Equations (36a) and (36b) reduce as follows:

$$\kappa_{eff}\,\delta\Theta_{(ab)} \approx \delta\overset{\circ}{G}_{ab} + O\Big(|\delta h|^2\Big), \tag{38a}$$

$$0 \approx O\Big(|\delta h|^2\Big). \tag{38b}$$

Equation (38b) for the antisymmetric part of the field equations is identically satisfied, while Equation (38a) shows that a variation $\delta\overset{\circ}{G}_{ab}$ associated with a perturbation is directly related to a variation of the energy-momentum tensor $\delta\Theta_{(ab)}$. This shows that the perturbations of Minkowski spacetime as defined in Section 3.1.1 for $f(T)$ teleparallel gravity follow the perturbative treatments of Minkowski spacetime in GR.

## 5. Effects of Perturbations and the Minkowski Spacetime Symmetries Conditions for Stability

### 5.1. Rotation/Boost Perturbation in a Minkowski Background

We would like to know if orthonormal coframe perturbations as expressed by Equation (32) lead to the stability of a pure Minkowski spacetime background. To achieve this goal, we will first test the stability for the rotation/boost perturbations as described in Equation (32). Secondly, we will also test the stability and its impact for a translated form of this Equation (32). We will finish by studying the effects of the trace, symmetric, and antisymmetric parts of perturbation, and their respective impacts on torsion and superpotential perturbations.

In fact, Equation (32) for the orthonormal gauge is exactly the rotation/boost perturbation in Minkowski spacetime. The perturbation is described as follows:

$$\delta h^{a}{}_{\mu} = \lambda^{a}{}_{b}\,h^{b}{}_{\mu}. \tag{39}$$

By substituting Equation (A18) inside Equation (38a), the field equation with the Equation (39) perturbation inside is exactly:

$$
\kappa_{eff}\, \delta\Theta_{(ab)} \;\approx\; \left(h_a{}^\mu h_b{}^\nu\right)\left[h_k{}^\alpha\, h^m{}_\mu\, \delta\overset{\circ k}{R}{}_{m\alpha\nu} - \frac{\eta^{cd}\,\eta_{ef}}{2}\left[h_c{}^\sigma\, h_d{}^\rho\, h^e{}_\mu\, h^f{}_\nu\right] h_k{}^\alpha\, h^m{}_\sigma\, \delta\overset{\circ k}{R}{}_{m\alpha\rho}\right]
$$
$$
+ O\left(|\delta h|^2\right),
$$
$$
0 \;\approx\; O\left(|\delta h|^2\right). \tag{40}
$$

Here, we obtain the perturbed FEs in terms of $\delta\overset{\circ k}{R}{}_{m\alpha\rho}$ and $h^a{}_\mu$. If we have that $\delta\overset{\circ k}{R}{}_{m\alpha\nu} \to 0$, then we obtain $\delta\Theta_{(ab)} \to 0$ for Equation (39), as is also required by GR and TEGR. We might also express Equation (40) in terms of $\lambda^a{}_b$, and we have shown that pure Minkowski spacetime is stable from the zero curvature criteria, as required by the teleparallel postulates.

From Equation (A8), and by substituting Equation (39), the torsion scalar perturbation $\delta T$ is expressed by Equation (A22) in Appendix C. This last equation can be summarized as:

$$
\delta T \to 0 \qquad \text{for } T^a{}_{\mu\nu} = \partial_\mu h^a{}_\nu - \partial_\nu h^a{}_\mu \to 0. \tag{41}
$$

From here, we obtain that the condition for $\delta T \to 0$ is described by the zero torsion tensor criteria $T^a{}_{\mu\nu} = 0$ relation as:

$$
\partial_\mu(h^a{}_\nu) \approx \partial_\nu\left(h^a{}_\mu\right) \tag{42}
$$

From Equation (A13), and by substituting Equation (39), the superpotential perturbation $\delta S_{ab}{}^\mu$ is expressed by Equation (A23) in Appendix C. This equation can be summarized as:

$$
\delta S_{ab}{}^\mu \to 0 \qquad \text{for } \delta T^a{}_{\mu\nu} = \partial_\mu\left(\lambda^a{}_c\, h^c{}_\nu\right) - \partial_\nu\left(\lambda^a{}_c\, h^c{}_\mu\right) \to 0. \tag{43}
$$

From this result, we obtain that the condition for $\delta S_{ab}{}^\mu \to 0$ is also described by the zero perturbed torsion tensor criteria $\delta T^a{}_{\mu\nu} = 0$ relation as:

$$
\partial_\mu\left(\lambda^a{}_b\, h^b{}_\nu\right) \approx \partial_\nu\left(\lambda^a{}_b\, h^b{}_\mu\right). \tag{44}
$$

Equation (44) (the zero perturbed torsion criteria) is complementary to Equation (42) (zero torsion criteria) for obtaining the limit $\delta S_{ab}{}^\mu \to 0$. We apply Equation (42) before applying Equation (44). From here, the Equations (42) and (44) are the **two fundamental symmetry conditions for Minkowski spacetime stability**.

If we set $\delta T \to 0$ and $\delta S_{ab}{}^\mu \to 0$ for Equations (36a) and (36b) for all zero torsion spacetimes, we still respect Equations (42) and (44), as for pure Minkowski spacetimes. Hence, the zero torsion tensor and zero perturbed torsion tensor criterions are still valid for all zero torsion spacetimes, Minkowski or not.

Even for the constant torsion spacetimes, by always setting $\delta T \to 0$ and $\delta S_{ab}{}^\mu \to 0$ inside Equations (35a) and (35b), we respect again Equations (42) and (44), as for the zero torsion scalar spacetimes. This is another generalization of the Minkowski spacetime result to a most general class of spacetimes as the constant torsion ones.

There are some other consequences for Minkowski spacetime on a proper frame. By applying the null covariant derivative criteria to Equation (39), we use Equation (A24) in the Appendix C result to obtain as a relation:

$$\delta\Gamma^{\rho}{}_{\nu\mu} = h_a{}^{\rho}\left[\partial_\mu\left(\lambda^a{}_b\,h^b{}_\nu\right) - \left(h_c{}^{\sigma}\,\partial_\mu\,h^c{}_\nu\right)\left(\lambda^a{}_b\,h^b{}_\sigma\right)\right], \tag{45}$$

where $\Gamma^{\rho}{}_{\nu\mu} = h_c{}^{\rho}\,\partial_\mu\,h^c{}_\nu$ is the Weitzenbock connection for a proper frame. For trivial coframes as $h^a{}_\mu = \delta^a{}_\mu = Diag[1,1,1,1]$, Equation (45) becomes:

$$\delta\Gamma^{\rho}{}_{\nu\mu} = h_a{}^{\rho}\left[\partial_\mu\left(\lambda^a{}_b\,h^b{}_\mu\right)\right] = \delta_a{}^{\rho}\,\partial_\mu(\lambda^a{}_b)\,\delta^b{}_\mu. \tag{46}$$

In the next subsection, we will study the effect of a translation applied to the perturbation described by Equation (39) on Equations (45) and (46). The goal is to know the effects of the perturbations on the Weitzenbock connection and its perturbation.

We can now see by the Equations (40)–(46) the effect of the perturbation described by Equation (39), maintaining the proper frame and respecting the $GL(4,\mathbb{R})$ invariance transformation. In addition, Equations (42) and (44) give the Minkowski spacetime stability conditions on proper frames for the perturbation described by Equation (39) [39–42].

*5.2. General Linear Perturbation in a Minkowski Background*

A more general perturbation scheme requires one to deal with the following general linear perturbation:

$$\delta h^a{}_\mu = \lambda^a{}_b\,h^b{}_\mu + \epsilon^a{}_\mu, \tag{47}$$

where $|\lambda^a{}_b|, |\epsilon^a{}_\mu| \ll 1$. We have here the transformation described by Equation (39), superposed with a translation in Minkowski tangent space.

For the Equation (47) perturbation, Equation (40) becomes as follows:

$$\begin{aligned}
\kappa_{eff}\,\delta\Theta_{(ab)} &\approx \left(h_a{}^\mu h_b{}^\nu\right)\left[h_k{}^\alpha\,h^m{}_\mu\,\delta\overset{\circ k}{R}{}_{m\alpha\nu} - \frac{\eta^{cd}\,\eta_{ef}}{2}\left[h_c{}^\sigma\,h_d{}^\rho\,h^e{}_\mu\,h^f{}_\nu\right]h_k{}^\alpha\,h^m{}_\sigma\,\delta\overset{\circ k}{R}{}_{m\alpha\rho}\right] \\
&\quad + O\left(|\delta h|^2\right), \\
0 &\approx O\left(|\delta h|^2\right).
\end{aligned} \tag{48}$$

Here, again we obtain the perturbed FEs in terms of $\delta\overset{\circ k}{R}{}_{m\alpha\rho}$ and $h^a{}_\mu$. As for Equation (40), $\delta\overset{\circ k}{R}{}_{m\alpha\nu} \to 0$, we still then obtain $\delta\Theta_{(ab)} \to 0$ for Equation (47), as is also required by GR and TEGR [39–42]. We might express Equation (48) in terms of $\lambda^a{}_b$ and $\epsilon^a{}_\mu$. Here again, we have shown that pure Minkowski spacetime is still stable from the zero curvature criteria, as required by teleparallel postulates.

From Equation (A8), and by substituting Equation (47), the torsion scalar perturbation $\delta T$ is expressed by Equation (A25) in Appendix C and can be summarized as:

$$\delta T \to 0 \qquad \text{for } T^a{}_{\mu\nu} = \partial_\mu\,h^a{}_\nu - \partial_\nu\,h^a{}_\mu \to 0. \tag{49}$$

The condition for $\delta T \to 0$ is still described by Equation (42) for the zero torsion tensor criteria $T^a{}_{\mu\nu} = 0$.

From Equation (A13), and by substituting Equation (47), the superpotential perturbation $\delta S_{ab}{}^\mu$ is expressed by Equation (A26) in Appendix C and can also be summarized as:

$$\delta S_{ab}{}^\mu \to 0. \tag{50}$$

Equation (50) is satisfied if we respect $\partial_a \epsilon_b{}^\mu = \partial_b \epsilon_a{}^\mu = 0$ (a constant translation condition for Equation (47)) and after applying the Equation (42) criteria. The condition for $\delta S_{ab}{}^\mu \to 0$ is still described by Equation (44) for the zero perturbed torsion tensor criteria $\delta T^a{}_{\mu\nu} = 0$, only if the constant translation criteria are respected as:

$$\partial_\mu \epsilon^a{}_\nu = \partial_\nu \epsilon^a{}_\mu = 0. \tag{51}$$

Hence, for the Equation (47) perturbation, we still respect Equations (42) and (44) as the two first symmetry conditions for Minkowski spacetime stability, but we must also respect Equation (51) before Equation (44). A simple translation does not affect these Equations (42) and (44) only if we respect Equation (51), and the translation term $\epsilon^a{}_\nu$ must be constant inside Equation (47). This constant translation criteria as expressed by Equation (51) is a **third symmetry condition for Minkowski spacetime stability**.

As for Equations (45) and (46), we apply the null covariant derivative criteria to Equation (47) and we obtain as a relation:

$$
\begin{aligned}
0 &= \partial_\mu \left( \lambda^a{}_b h^b{}_\nu + \epsilon^a{}_\nu \right) - \left( h_c{}^\rho \partial_\mu h^c{}_\nu \right) \left( \lambda^a{}_b h^b{}_\rho + \epsilon^a{}_\rho \right) - \delta\Gamma^\rho{}_{\nu\mu} h^a{}_\rho \\
&\Rightarrow \delta\Gamma^\rho{}_{\nu\mu} = h_a{}^\rho \left[ \partial_\mu \left( \lambda^a{}_b h^b{}_\nu \right) - \left( h_c{}^\sigma \partial_\mu h^c{}_\nu \right) \left( \lambda^a{}_b h^b{}_\sigma + \epsilon^a{}_\sigma \right) \right].
\end{aligned} \tag{52}
$$

where $\Gamma^\sigma{}_{\nu\mu} = h_c{}^\sigma \partial_\mu h^c{}_\nu$ is the Weitzenbock connection for a proper frame and $\partial_\mu \epsilon^a{}_\nu = 0$ because of constant translation. Equation (52) is slightly different from Equation (45) according to the term $-\left( h_c{}^\sigma \partial_\mu h^c{}_\nu \right) \epsilon^a{}_\sigma$. For non-trivial coframes (i.e., $\partial_\mu h^c{}_\nu \neq 0$), Equation (52) is not invariant under Equation (51). For trivial coframes (i.e., $\partial_\mu h^c{}_\nu = 0$), Equation (52) becomes exactly Equation (46), as for the perturbation described by Equation (39). From this result, we now respect the constant coframe criteria as (or null Weitzenbock connection $\Gamma^\rho{}_{\nu\mu} = 0$ criteria):

$$\partial_\mu h^c{}_\nu = 0. \tag{53}$$

With Equation (53), we also satisfy the invariance under Equation (51), the constant translation criteria for the Weitzenbock connection perturbation. Hence, Equations (45) and (52) show that the Weitzenbock connection perturbation $\delta\Gamma^\rho{}_{\nu\mu}$ is invariant only if we respect Equation (53), the constant coframe criteria. This criteria, as expressed by Equation (53), is a **fourth symmetry condition for Minkowski spacetime stability**.

Now, Equations (48)–(53) generalize Equations (40)–(46) by applying a constant translation $\epsilon^a{}_\nu$ to the linear transformation described by Equation (39), which maintains the proper frame and the invariance under the $GL(4, \mathbb{R})$ transformation. By respecting Equation (51), the constant translation criteria, we still respect Equations (42) and (44) for Equation (47), and this generalization shows that Minkowski spacetime and all zero torsion spacetimes are stable everytime [39–42]. However, Equations (45) and (52), both giving Equation (46), show that the Weitzenbock connection perturbation $\delta\Gamma^\rho{}_{\nu\mu}$ is invariant only if we work with constant or trivial coframes respecting Equation (53).

*5.3. Perturbations on Trivial Coframes by Each Part of the Perturbation*

Before properly dealing with more complex cases of coframes, it is imperative to deal with perturbations on the trivial coframe. This coframe is defined as follows:

$$h^a{}_\mu = \delta^a{}_\mu = Diag[1,\ 1,\ 1,\ 1]. \tag{54}$$

The coframe described by Equation (54) is defined in the orthonormal gauge. This Equation (54) respects Equation (53), the fourth symmetry condition for Minkowski spacetime stability. From there, we will study the following general perturbations which will be applied to Equation (54) in terms of $\lambda^a{}_b$ and respecting Equations (42) and (44), and if necessary, Equation (51). In addition, we will compare with another recent similar study on

so-called "cosmological" perturbations in order to better situate the results for Minkowski spacetime for a scale factor of 1 [24]. Their $\lambda^a{}_b$ equivalent matrix is expressed as:

$$(\lambda^a{}_b)_{Golov} = \begin{bmatrix} \phi & \partial_a\,\xi + v_a \\ \partial_i\,\beta + u_i & \left[ -\psi\,\delta^a_j + \partial^2_{a\,j}\sigma + \epsilon_{ajk}\,(\partial_k\,s + w_k) + \partial_j\,c_a + \frac{h_{aj}}{2} \right] \end{bmatrix}, \tag{55}$$

where we must respect the constraints $\partial^a\,v_a = 0$, $\partial^k\,w_k = 0$, $\partial^i\,u_i = 0$, and $\partial^a\,c_a = 0$, and the tensorial part is also traceless.

5.3.1. Trace

We have first as $\lambda^a{}_b$ for a full trace perturbation:

$$(\lambda^a{}_b)_{Trace} = \lambda = Trace[Diag[a_{00}, a_{11}, a_{22}, a_{33}]] = a_{00} + a_{11} + a_{22} + a_{33}. \tag{56}$$

Equation (39) will be exactly $\left(\delta h^a{}_\mu\right)_{Trace} = \frac{\lambda}{4}\delta^a{}_\mu$, and by setting $h^a{}_\mu = \delta^a{}_\mu$, Equations (41) and (43) are:

$$\delta T \approx O\left(|\delta h|^2\right) \to 0, \tag{57}$$

which respects Equation (42) and

$$\delta S_{ab}{}^\mu \to \left[ \frac{1}{8}\left(\partial_a\left(\lambda\,\delta_b{}^\mu\right) - \partial_b\left(\lambda\,\delta_a{}^\mu\right)\right) + \frac{1}{4}\left(\partial_b\left(\lambda\,\delta^\mu_a\right) - \partial_a\left(\lambda\,\delta^c_b\,h^\mu_c\right)\right) \right.$$

$$\left. - \frac{1}{4}\delta^\mu_b\,\delta^c_\rho\left[\partial_c\left(\lambda\,\delta_a{}^\rho\right) - \partial_a\left(\lambda\,\delta_c{}^\rho\right)\right] + \frac{1}{4}\delta^\mu_a\,\delta^c_\rho\left[\partial_c\left(\lambda\,\delta_b{}^\rho\right) - \partial_b\left(\lambda\,\delta_c{}^\rho\right)\right] \right]$$

$$+ O\left(|\delta h|^2\right) \qquad \text{by applying Equation (42) (the zero torsion criteria).}$$

$$\to 0 \qquad\qquad \text{for } \delta T^a{}_{\mu\nu} = \partial_\mu\,(\lambda)\,\delta^a{}_\nu - \partial_\nu\,(\lambda)\,\delta^a{}_\mu \to 0. \tag{58}$$

Equation (44) will be expressed as:

$$\partial_\mu\,(\lambda)\,\delta^a{}_\nu \approx \partial_\nu\,(\lambda)\,\delta^a{}_\mu. \tag{59}$$

By comparing with Equation (55), we obtain the following equations for the rectangular coordinates [24]:

- Equation (56) becomes:

$$(\lambda^a{}_b)_{Trace\,Golov} = \lambda_{Golov} = \phi - \psi + \partial^2\,\sigma + \frac{h}{2}, \tag{60}$$

where $\epsilon_{ajk} = 0$ because $a = j$ and $h = Trace(h_{aj})$.

- From Equation (56), we obtain as the supplementary constraints:

$$\partial_a\,\xi + v_a = 0 \quad \text{and} \quad \partial_i\,\beta + u_i = 0 \tag{61}$$

- Equation (59) will be expressed in terms of Equations (60) and (61):

$$\partial_\mu\,(\lambda_{Golov})\,\delta^a{}_\nu \approx \partial_\nu\,(\lambda_{Golov})\,\delta^a{}_\mu$$

$$\partial_\mu\left(\phi - \psi + \partial^2\,\sigma + \frac{h}{2}\right)\delta^a{}_\nu \approx \partial_\nu\left(\phi - \psi + \partial^2\,\sigma + \frac{h}{2}\right)\delta^a{}_\mu. \tag{62}$$

### 5.3.2. Full Symmetric Perturbation

For the perfect symmetric perturbation, we have as the $\lambda^a_b$ perturbation with null diagonal components:

$$(\lambda^a_b)_{Sym} = \tilde{\lambda}^a_b = \begin{bmatrix} 0 & b_{10} & b_{20} & b_{30} \\ b_{10} & 0 & b_{12} & b_{13} \\ b_{20} & b_{12} & 0 & b_{23} \\ b_{30} & b_{13} & b_{23} & 0 \end{bmatrix}. \tag{63}$$

Equation (39) will be exactly $\left(\delta h^a_{\ \mu}\right)_{Sym} = \tilde{\lambda}^a_b \, \delta^b_{\ \mu}$, and by setting $h^a_{\ \mu} = \delta^a_{\ \mu}$, Equation (41) is still expressed by Equation (57), respecting the Equations (42) and (43):

$$\delta S_{ab}{}^\mu = \left[ \frac{1}{2} \left( \partial_a \left( \tilde{\lambda}^c_b \, \delta^\mu_c \right) - \partial_b \left( \tilde{\lambda}^c_a \, \delta^\mu_c \right) \right) + \left( \partial_b \left( \tilde{\lambda}^c_a \, \delta^\mu_c \right) - \partial_a \left( \tilde{\lambda}^c_b \, \delta^\mu_c \right) \right) \right.$$

$$\left. - \delta^\mu_b \, \delta^c_\rho \left[ \partial_c \left( \tilde{\lambda}^f_a \, \delta^\rho_f \right) - \partial_a \left( \tilde{\lambda}^f_c \, \delta^\rho_f \right) \right] + \delta^\mu_a \, \delta^c_\rho \left[ \partial_c \left( \tilde{\lambda}^f_b \, \delta^\rho_f \right) - \partial_b \left( \tilde{\lambda}^f_c \, \delta^\rho_f \right) \right] \right]$$

$$+ O\left( |\delta h|^2 \right) \qquad \text{by applying Equation (42) (the zero torsion criteria).}$$

$$\to 0 \qquad \qquad \text{for } \delta T^a_{\mu\nu} = \partial_\mu \left( \lambda^a_c \right) \delta^c_{\ \nu} - \partial_\nu \left( \lambda^a_c \right), \delta^c_{\ \mu} \to 0. \tag{64}$$

Equation (44) will be expressed as:

$$\partial_\mu \left( \tilde{\lambda}^a_c \right) \delta^c_{\ \nu} \approx \partial_\nu \left( \tilde{\lambda}^a_c \right), \delta^c_{\ \mu}. \tag{65}$$

By comparing with Equation (55) again, we obtain the following equations for the rectangular coordinates [24]:

- Equation (63) becomes:

$$\begin{aligned} (\lambda^a_b)_{Sym\,Golov} &= \left( \tilde{\lambda}^a_b \right)_{Golov} \\ &= \begin{bmatrix} 0 & \partial_a \, \xi + v_a \\ \partial_a \, \xi + v_a & \left[ \partial^2_{aj} \sigma + \partial_j \, c_a + \frac{h_{aj}}{2} \right] \end{bmatrix}, \end{aligned} \tag{66}$$

where $a \neq j \neq k$, $\epsilon_{ajk} \left( \partial_k s + w_k \right) = 0$ and $\partial_a \, \xi + v_a = \partial_i \, \beta + u_i$ because we have a symmetric perturbation. As a supplement, we deduce that $\phi = 0$ and $\psi = 0$ for Equation (63), because of the null diagonal components.

- The Equation (65) components will be expressed in terms of Equation (66):

$$\partial_\mu \left( \partial_a \, \xi + v_a \right) \delta^a_{\ \nu} \approx \partial_\nu \left( \partial_a \, \xi + v_a \right) \delta^a_{\ \mu}$$

$$\partial_\mu \left( \partial^2_{aj} \sigma + \partial_j \, c_a + \frac{h_{aj}}{2} \right) \delta^a_{\ \nu} \approx \partial_\nu \left( \partial^2_{aj} \sigma + \partial_j \, c_a + \frac{h_{aj}}{2} \right) \delta^a_{\ \mu} \tag{67}$$

### 5.3.3. Full Antisymmetric Perturbation

For the full antisymmetric perturbation, we have as the $\lambda^a_b$ perturbation with null diagonal components:

$$(\lambda^a_b)_{AntiSym} = \bar{\lambda}^a_b = \begin{bmatrix} 0 & b_{10} & b_{20} & b_{30} \\ -b_{10} & 0 & b_{12} & b_{13} \\ -b_{20} & -b_{12} & 0 & b_{23} \\ -b_{30} & -b_{13} & -b_{23} & 0 \end{bmatrix}. \tag{68}$$

Equation (39) will be exactly $\left(\delta h^a{}_\mu\right)_{AntiSym} = \bar{\lambda}^a{}_b\,\delta^b{}_\mu$, and by setting $h^a{}_\mu = \delta^a{}_\mu$, Equation (41) is still expressed by Equation (57), respecting Equations (42) and (43):

$$
\begin{aligned}
\delta S_{ab}{}^\mu = & \left[ \frac{1}{2}\left(\partial_a\left(\bar{\lambda}_b{}^c\,\delta_c{}^\mu\right) - \partial_b\left(\bar{\lambda}_a{}^c\,\delta_c{}^\mu\right)\right) + \left(\partial_b\left(\bar{\lambda}^c{}_a\,\delta^\mu{}_c\right) - \partial_a\left(\bar{\lambda}^c{}_b\,\delta^\mu{}_c\right)\right) \right. \\
& \left. - \delta^\mu{}_b\,\delta^c{}_\rho\left[\partial_c\left(\bar{\lambda}_a{}^f\delta_f{}^\rho\right) - \partial_a\left(\bar{\lambda}_c{}^f\delta_f{}^\rho\right)\right] + \delta^\mu{}_a\,\delta^c{}_\rho\left[\partial_c\left(\bar{\lambda}_b{}^f\delta_f{}^\rho\right) - \partial_b\left(\bar{\lambda}_c{}^f\delta_f{}^\rho\right)\right] \right]
\end{aligned}
$$

$$
+ O\left(|\delta h|^2\right) \qquad \text{by applying Equation (42) (the zero torsion criteria).}
$$

$$
\to 0 \qquad\qquad \text{for } \delta T^a_{\mu\nu} = \partial_\mu\left(\bar{\lambda}^a{}_c\right)\delta^c{}_\nu - \partial_\nu\left(\bar{\lambda}^a{}_c\right)\delta^c{}_\mu \to 0. \tag{69}
$$

Equation (44) will be expressed as:

$$
\partial_\mu\left(\bar{\lambda}^a{}_c\right)\delta^c{}_\nu \approx \partial_\nu\left(\bar{\lambda}^a{}_c\right)\delta^c{}_\mu. \tag{70}
$$

By still comparing with Equation (55), we obtain the following equations for the rectangular coordinates [24]:

- Equation (68) becomes:

$$
\begin{aligned}
\left(\lambda^a{}_b\right)_{AntiSym\,Golov} &= \left(\bar{\lambda}^a{}_b\right)_{Golov} \\
&= \begin{bmatrix} 0 & \partial_a\,\xi + v_a \\ -(\partial_a\,\xi + v_a) & \left[\epsilon_{ajk}\left(\partial_k\,s + w_k\right) + \partial_j\,c_a + \frac{h_{aj}}{2}\right] \end{bmatrix}
\end{aligned} \tag{71}
$$

where $a \neq j \neq k$, $\partial^2_{a\,j}\sigma = -\partial^2_{j\,a}\sigma = 0$, $\partial_j\,c_a = -\partial_a\,c_j$, $h_{aj} = -h_{ja}$, and $\partial_a\,\xi + v_a = -(\partial_i\,\beta + u_i)$ because we have an antisymmetric perturbation. We deduce again that $\phi = 0$ and $\psi = 0$ for Equation (68), because of the null diagonal components.

- The Equation (70) components will be expressed in terms of Equation (71):

$$
\begin{aligned}
\partial_\mu\left(\partial_a\,\xi + v_a\right)\delta^a{}_\nu &\approx \partial_\nu\left(\partial_a\,\xi + v_a\right)\delta^a{}_\mu \\
\partial_\mu\left(\epsilon_{ajk}\left(\partial_k\,s + w_k\right) + \partial_j\,c_a + \frac{h_{aj}}{2}\right)\delta^a{}_\nu &\approx \partial_\nu\left(\epsilon_{ajk}\left(\partial_k\,s + w_k\right) + \partial_j\,c_a + \frac{h_{aj}}{2}\right)\delta^a{}_\mu
\end{aligned}
$$

$$
\tag{72}
$$

### 5.3.4. A Mixed Situation and Minkowski Spacetime

Here, we will treat the most general case. It is the combination of the three previous sorts, as:

$$
\begin{aligned}
\left(\lambda^a{}_b\right)_{Mixed} = \lambda^a{}_b &= \frac{\delta^a{}_b}{4}\left(\lambda^a{}_b\right)_{Trace} + \left(\lambda^a{}_b\right)_{Sym} + \left(\lambda^a{}_b\right)_{AntiSym}, \\
&= \frac{\lambda}{4}\delta^a{}_b + \tilde{\lambda}^a{}_b + \bar{\lambda}^a{}_b.
\end{aligned} \tag{73}
$$

In general, we always obtain for Equation (73) that $\left(\lambda^a{}_b\right)_{Mixed}$ is exactly Equation (55) when we compare it to the linear parametrization of ref [24]. Then, we obtain as the

components of Equation (44) the most general relations for perturbation in the Minkowski background as:

$$\partial_\mu \phi \delta^c{}_\nu \approx \partial_\nu \phi, \delta^c{}_\mu \tag{74a}$$

$$\partial_\mu (\partial_a \xi + v_a) \delta^c{}_\nu \approx \partial_\nu (\partial_a \xi + v_a), \delta^c{}_\mu \tag{74b}$$

$$\partial_\mu (\partial_i \beta + u_i) \delta^c{}_\nu \approx \partial_\nu (\partial_i \beta + u_i), \delta^c{}_\mu \tag{74c}$$

$$\partial_\mu \left[ -\psi \delta^a_j + \partial^2_{aj}\sigma + \epsilon_{ajk}(\partial_k s + w_k) + \partial_j c_a + \frac{h_{aj}}{2} \right] \delta^c{}_\nu$$

$$\approx \partial_\nu \left[ -\psi \delta^a_j + \partial^2_{aj}\sigma + \epsilon_{ajk}(\partial_k s + w_k) + \partial_j c_a + \frac{h_{aj}}{2} \right] \delta^c{}_\mu. \tag{74d}$$

Equation (39) will be exactly $\left( \delta h^a{}_\mu \right)_{Mixed} = \lambda^a{}_b \, \delta^b{}_\mu$, and we exactly obtain Equations (41) and (43) by respecting Equations (42) and (44) via superposition. In Equation (73), the first two terms (Trace and Symmetric terms) represent the symmetric part of $\left( \lambda^a{}_b \right)_{Mixed}$, and the last term (Antisymmetric term) represents the Antisymmetric part of $\left( \lambda^a{}_b \right)_{Mixed}$. For every case, we satisfy the Equations (42), (44), (51), and (53) in the supplement of the energy-momentum stability from $\delta \overset{\circ}{R}{}^k{}_{m\alpha\nu} \to 0$, leading to $\delta\Theta_{(ab)} \to 0$ [39–42].

For the trivial coframe cases expressed by Equation (54), we verify the energy-momentum stability by Equation (40); and the four other symmetries conditions stated by Equations (42), (44), (51), and (53) are all satisfied. The Minkowski spacetime is stable with these four symmetry conditions. From these considerations for pure Minkowski spacetime, we have shown that $\delta\Theta_{(ab)} \to 0$ by Equations (40) and (48) when all of the perturbed quantities proceed to zero. From this, we must absolutely have $\Theta_{(ab)} = 0$ when we are in a pure vacuum: the full absence of a gravitational source.

## 6. Discussion and Conclusions

The purpose of this paper is to clarify the meaning of Minkowski and constant scalar torsion geometries within a teleparallel gravity framework. A perturbation scheme is developed, which is general and applicable to all possible teleparallel spacetimes that respect the Null Curvature and Null Non-Metricity postulates. The perturbation scheme is then applied to different constant torsion scalar scenarios, with a particular emphasis on perturbations of the teleparallel Minkowksi spacetimes.

We obtained in Section 4 the perturbed field equations (perturbed FEs) in terms of the perturbed torsion scalar $\delta T$ and perturbed superpotential $\delta S_{ab}{}^\mu$. These two quantities are themselves dependent on the coframe perturbation $\delta h^a{}_\mu$. The perturbed field equations make it possible to relate these perturbed quantities to the perturbation of the energy-momentum $\delta\Theta_{(ab)}$. This is analogous to the field equations for the non-perturbed quantities and how they relate to the physical quantities in the Energy-Momentum $\Theta_{(ab)}$.

In Section 5, we look at the field Equations (40) and (48) when the curvature perturbation criteria $\delta \overset{\circ}{R}{}^k{}_{m\alpha\nu}$ proceeds to zero, and we observe that the energy-momentum perturbation $\delta\Theta_{(ab)}$ also goes to zero, as in GR. In GR, it is known that a curvature perturbation leads to an energy-momentum perturbation. We show that the same thing occurs for the teleparallel Minkowski spacetime with Equations (40) and (48).

Then, we obtain via the null torsion tensor and the null perturbed torsion tensor criteria as defined by Equations (42) and (44) that the torsion scalar perturbation $\delta T$ and superpotential perturbation $\delta S_{ab}{}^\mu$ go to zero for pure Minkowski spacetime when we use the Equation (39) perturbation (boost/rotation perturbation). These Equations (42) and (44) are the two first fundamental Minkowski spacetime stability conditions on proper frames. However, if we use the more general linear perturbation as defined by Equation (47), we need to respect the constant translation criteria as defined by Equation (51) in order for the superpotential perturbation $\delta S_{ab}{}^\mu$ to proceed to zero. This is a third Minkowski spacetime stability condition for the proper frames to respect for the Equation (47) perturbation. In

this way, by respecting Equation (51), we then respect the Equations (42) and (44), as for the Equation (39) perturbation.

Another consequence from the Equation (47) perturbation is about the Weitzenbock connection perturbation $\delta\Gamma^{\rho}{}_{\nu\mu}$. Equations (45) and (52) have shown that we need to respect the constant coframe criteria as defined by Equation (53). Equation (53) is a fourth Minkowski spacetime stability condition for proper frames to respect for the Equation (47) perturbation, allowing for the invariance for the Weitzenbock connection perturbation.

To generalize, these steps applied for the Minkowski spacetime, given these stability criteria, can also be applied for null torsion scalar spacetimes, as well as the constant torsion scalar spacetimes. Indeed, with the analysis made in Sections 5.1 and 5.2, and the stability criteria obtained for the Minkowski spacetime, Equations (36a) and (36b) for the null torsion scalar spacetimes make it possible to generalize these treatments, and in the end to obtain the same stability criteria, which are Equations (42) and (44), and if necessary, Equations (51) and (53). This is also the case for the constant torsion scalar spacetimes described by the Equations (35a) and (35b) if we take the limits $\delta T \to 0$ and $\delta S_{ab}{}^{\mu} \to 0$, as for the Minkowski and null torsion scalar spacetimes.

One can expand upon the work here on perturbations in covariant teleparallel gravity to more general teleparallel spacetimes and to broader classes of teleparallel gravity theories. For example, in the case of static spherically symmetric teleparallel spacetimes [43,44] in which the torsion scalar is not constant, what is the stability of the static spherically symmetric solution? Further, this perturbation scheme can also be applied to cosmological geometries in $f(T)$ teleparallel gravity [21], thereby enhancing the previous work of [24]. Additionally, one can also look at perturbations in other non-$f(T)$ teleparallel gravity theories.

The current analysis could also bring some light to a couple of unresolved challenges in teleparallel gravity. The first challenge concerns the number of degrees of freedom (DOF) in 4-dimensional $f(T)$ teleparallel gravity [45–48]. In [45], the authors employ a Hamiltonian analysis to determine that $f(T)$ teleparallel gravity has three extra DOF when compared to GR. Unfortunately, it appears that the analysis is flawed, in that it is not general, for they assumed a diagonal metric to reach some of their conclusions. Later, Ferraro and Guzmán [46] made an argument that the number of extra DOF is 1. However, the analysis appears to be somewhat incomplete and only applicable to teleparallel gravity geometries in which the torsion scalar is constant [48]. More recently, the authors of [47] go through a Hamiltonian analysis to conclude that the number of extra DOF is 3. A couple of challenges in their results have been identified in [48]. Obviously, this is still an unresolved problem which requires further investigation. Another unresolved complex physical problem is the strong coupling of teleparallel perturbations. This physical problem occurs as one approaches the Planck scale where the quantum field effects become non-negligible, particularly for second-order perturbations and higher. At these scales, the kinetic energy part will become dominant when compared to the gravity and background parts. This strong coupling issue with teleparallel perturbations needs further development and understanding within the covariant $f(T)$ teleparallel-gravity framework.

Here, with the material developed in this present paper, we have a more complete perturbation framework that is suitable for use in teleparallel gravity, and the toolkit needed for studying several and more complex problems in teleparallel gravity.

**Author Contributions:** Conceptualization, A.L. and R.J.v.d.H.; methodology, A.L. and R.J.v.d.H.; formal analysis, A.L. and R.J.v.d.H.; investigation, A.L.; resources, A.L. and R.J.v.d.H.; writing—original draft preparation, A.L.; writing—review and editing, A.L. and R.J.v.d.H. All authors have read and agreed to the published version of the manuscript.

**Funding:** R.J.v.d.H. is supported by the Natural Sciences and Engineering Research Council of Canada, and by the W.F. James Chair of Studies in the Pure and Applied Sciences at St.F.X. A.L. is supported by an AARMS fellowship.

**Conflicts of Interest:** The authors declare no conflict of interest.

## Abbreviations

The following abbreviations are used in this manuscript:

FE      Field Equation
GR      General Relativity
TEGR      Teleparallel Equivalent of General Relativity
DOF      Degrees of Freedom

## Appendix A. Perturbed Physical Quantities in Teleparallel Theories

To complete the analysis of Teleparallel theories and geometries, we want to perturb various physical quantities that may be involved. As explained in Section 4.1, we are able to always consider perturbations of the co-frame only within a proper orthonormal gauge.

$$
\begin{aligned}
\hat{h}'^a{}_\mu &= h^a{}_\mu + \delta h^a{}_\mu, & \text{(A1a)} \\
\hat{\omega}'^a{}_{b\mu} &= 0, & \text{(A1b)} \\
\hat{g}'_{ab} &= \eta_{ab}, & \text{(A1c)}
\end{aligned}
$$

where $\delta h^a = \delta h^a{}_\mu\, dx^\mu = \lambda^a{}_b h^b$. Here, we apply the coframe perturbations to the main physical and geometrical quantities involved in Teleparallel Gravity.

1.  The inverse coframe perturbation $\delta h_a{}^\mu$:

$$
\begin{aligned}
h_a{}^\mu + \delta h_a{}^\mu &= h_a{}^\mu + \left[\lambda^b{}_a\right]^{-1} h_a{}^\mu, \\
&= h_a{}^\mu + \lambda_a{}^b h_a{}^\mu, \\
\Rightarrow \delta h_a{}^\mu &= \lambda_a{}^b h_a{}^\mu
\end{aligned}
\qquad \text{(A2)}
$$

2.  Determinant of the co-frame $h = \mathrm{Det}(h^a{}_\mu)$:

$$
\begin{aligned}
h + \delta h &= \mathrm{Det}(h^a{}_\mu + \delta h^a{}_\mu) \\
&\approx h + \mathrm{Det}(\lambda^a{}_b h^b{}_\mu) = h + \lambda\, h \\
\Rightarrow \delta h &\approx \lambda\, h
\end{aligned}
\qquad \text{(A3)}
$$

where $\lambda = \mathrm{Det}(\lambda^a{}_b) \ll 1$ and $\mathrm{Det}(\delta h^a{}_\mu) = \mathrm{Det}(\lambda^a{}_b h^b{}_\mu) = \lambda\, h$.

3.  Metric tensor $g_{\mu\nu}$:

$$
\begin{aligned}
g_{\mu\nu} + \delta g_{\mu\nu} &= \eta_{ab}\left[h^a{}_\mu + \delta h^a{}_\mu\right]\left[h^b{}_\nu + \delta h^b{}_\nu\right], \\
&\approx g_{\mu\nu} + \eta_{ab}\left[\delta h^a{}_\mu h^b{}_\nu + h^a{}_\mu \delta h^b{}_\nu\right] + O\left(|\delta h|^2\right), \\
\Rightarrow \delta g_{\mu\nu} &\approx \eta_{ab}\left[\delta h^a{}_\mu h^b{}_\nu + h^a{}_\mu \delta h^b{}_\nu\right] + O\left(|\delta h|^2\right).
\end{aligned}
\qquad \text{(A4)}
$$

4.  Torsion tensor $T^a{}_{\mu\nu}$ and $T^\rho{}_{\mu\nu}$:

$$
\begin{aligned}
T^a{}_{\mu\nu} + \delta T^a{}_{\mu\nu} &= \partial_\mu h^a{}_\nu + \partial_\mu(\delta h^a{}_\nu) - \partial_\nu h^a{}_\mu + \partial_\nu\left(\delta h^a{}_\mu\right) \\
&\approx T^a{}_{\mu\nu} + \left[\partial_\mu(\delta h^a{}_\nu) - \partial_\nu\left(\delta h^a{}_\mu\right)\right] + O\left(|\delta h|^2\right) \\
\Rightarrow \delta T^a{}_{\mu\nu} &\approx \left[\partial_\mu(\delta h^a{}_\nu) - \partial_\nu\left(\delta h^a{}_\mu\right)\right] + O\left(|\delta h|^2\right)
\end{aligned}
\qquad \text{(A5)}
$$

If we also have that $T^\rho{}_{\mu\nu} = h_a{}^\rho\, T^a{}_{\mu\nu}$, then:

$$
\begin{aligned}
T^{\rho}{}_{\mu\nu} + \delta T^{\rho}{}_{\mu\nu} &= \left( h_a{}^{\rho} + \delta h_a{}^{\rho} \right)\left( T^a{}_{\mu\nu} + \delta T^a{}_{\mu\nu} \right) \\
&\approx T^{\rho}{}_{\mu\nu} + \delta h_a{}^{\rho}\, T^a{}_{\mu\nu} + h_a{}^{\rho}\, \delta T^a{}_{\mu\nu} + O\left( |\delta h|^2 \right) \\
\Rightarrow \delta T^{\rho}{}_{\mu\nu} &\approx \delta h_a{}^{\rho}\left[ \partial_\mu h^a{}_\nu - \partial_\nu h^a{}_\mu \right] + h_a{}^{\rho}\left[ \partial_\mu(\delta h^a{}_\nu) - \partial_\nu\left( \delta h^a{}_\mu \right) \right] + O\left( |\delta h|^2 \right)
\end{aligned}
$$

(A6)

5. Torsion scalar $T$:

$$
\begin{aligned}
T + \delta T &= \frac{1}{4}\left( T^a{}_{\mu\nu} + \delta T^a{}_{\mu\nu} \right)\left( T_a{}^{\mu\nu} + \delta T_a{}^{\mu\nu} \right) + \frac{1}{2}\left( T^a{}_{\mu\nu} + \delta T^a{}_{\mu\nu} \right)\left( T^{\nu\mu}{}_a + \delta T^{\nu\mu}{}_a \right) \\
&\quad - \left( T^{\nu}{}_{\mu\nu} + \delta T^{\nu}{}_{\mu\nu} + \right)\left( T^{\rho\mu}{}_\rho + \delta T^{\rho\mu}{}_\rho \right) \\
&= T + \frac{1}{4}\left( \delta T^a{}_{\mu\nu} T_a{}^{\mu\nu} + T^a{}_{\mu\nu}\delta T_a{}^{\mu\nu} \right) + \frac{1}{2}\left( \delta T^a{}_{\mu\nu} T^{\nu\mu}{}_a + T^a{}_{\mu\nu}\delta T^{\nu\mu}{}_a \right) \\
&\quad - \left( \delta T^{\nu}{}_{\mu\nu} T^{\rho\mu}{}_\rho + T^{\nu}{}_{\mu\nu}\delta T^{\rho\mu}{}_\rho \right) + O\left( |\delta h|^2 \right) \\
\Rightarrow \delta T &= \frac{1}{4}\left( \delta T^a{}_{\mu\nu} T_a{}^{\mu\nu} + T^a{}_{\mu\nu}\delta T_a{}^{\mu\nu} \right) + \frac{1}{2}\left( \delta T^a{}_{\mu\nu} T^{\nu\mu}{}_a + T^a{}_{\mu\nu}\delta T^{\nu\mu}{}_a \right) \\
&\quad - \left( \delta T^{\nu}{}_{\mu\nu} T^{\rho\mu}{}_\rho + T^{\nu}{}_{\mu\nu}\delta T^{\rho\mu}{}_\rho \right) + O\left( |\delta h|^2 \right)
\end{aligned}
$$

(A7)

In terms of Equations (A5) and (A6), Equation (A7) becomes as:

$$
\begin{aligned}
\delta T = \frac{1}{4}&\left[ \left( \partial_\mu(\delta h^a{}_\nu) - \partial_\nu\left( \delta h^a{}_\mu \right) \right)\left( \partial^\mu h_a{}^\nu - \partial^\nu h_a{}^\mu \right) + \left( \partial_\mu h^a{}_\nu - \partial_\nu h^a{}_\mu \right) \right. \\
&\left. \times \left( \partial^\mu\left( \delta h_a{}^\nu \right) - \partial^\nu\left( \delta h_a{}^\mu \right) \right) \right] + \frac{1}{2}\left[ \left( \partial_\mu(\delta h^a{}_\nu) - \partial_\nu\left( \delta h^a{}_\mu \right) \right)\left( \partial^\nu h^\mu{}_a - \partial^\mu h^\nu{}_a \right) \right. \\
&\left. + \left( \partial_\mu h^a{}_\nu - \partial_\nu h^a{}_\mu \right)\left( \partial^\nu\left( \delta h^\mu{}_a \right) - \partial^\mu\left( \delta h^\nu{}_a \right) \right) \right] \\
&- \left[ \left( \delta h_a{}^\nu\left[ \partial_\mu h^a{}_\nu - \partial_\nu h^a{}_\mu \right] + h_a{}^\nu\left[ \partial_\mu(\delta h^a{}_\nu) - \partial_\nu\left( \delta h^a{}_\mu \right) \right] \right)\left( h^a{}_\rho\left( \partial^\rho h^\mu{}_a - \partial^\mu h^\rho{}_a \right) \right) \right. \\
&\left. + \left( h_a{}^\nu\left[ \partial_\mu h^a{}_\nu - \partial_\nu h^a{}_\mu \right] \right)\left( \delta h^a{}_\rho\left( \partial^\rho h^\mu{}_a - \partial^\mu h^\rho{}_a \right) + h^a{}_\rho\left( \partial^\rho\left( \delta h^\mu{}_a \right) - \partial^\mu\left( \delta h^\rho{}_a \right) \right) \right) \right] \\
&+ O\left( |\delta h|^2 \right).
\end{aligned}
$$

(A8)

6. Lagrangian density $\mathcal{L}_{Grav}$:

$$
\begin{aligned}
\mathcal{L}_{Grav} + \delta\mathcal{L}_{Grav} &= \frac{1}{2\kappa}(h + \delta h)\, f(T + \delta T), \\
&\approx \mathcal{L}_{Grav} + \frac{1}{2\kappa}[\delta h\, f(T) + h\, f_T(T)\, \delta T] + O\left( |\delta h|^2 \right), \\
\Rightarrow \delta\mathcal{L}_{Grav} &\approx \frac{1}{2\kappa}[\delta h\, f(T) + h\, f_T(T)\, \delta T] + O\left( |\delta h|^2 \right).
\end{aligned}
$$

(A9)

7. Sum of the Torsion and Ricci Curvature scalar $\overset{\circ}{R} + T$: Here, $\overset{\circ}{R}$ is the Ricci scalar computed from the Levi-Civita connection.

$$\delta(\overset{\circ}{R} + T) = \delta\left[\frac{2}{h}\,\delta_\mu\left(h\,T^{\nu\mu}{}_\nu\right)\right] = 2\left[\delta\left(\frac{1}{h}\right)\delta_\mu\left(h\,T^{\nu\mu}{}_\nu\right) + \frac{1}{h}\delta_\mu\left[\delta\left(h\,T^{\nu\mu}{}_\nu\right)\right]\right]$$

$$\approx \frac{2}{h}\left[-\frac{\delta h}{h}(\delta_\mu h)\,T^{\nu\mu}{}_\nu + (\delta_\mu(\delta h))\,T^{\nu\mu}{}_\nu + (\delta_\mu h)\,\delta T^{\nu\mu}{}_\nu + h\,\delta_\mu\left(\delta T^{\nu\mu}{}_\nu\right)\right]$$

$$+\,O\left(|\delta h|^2\right) \tag{A10}$$

By using Equation (A6), Equation (A10) becomes as:

$$\delta(\overset{\circ}{R} + T) \approx \frac{2}{h}\left[-\frac{\delta h}{h}(\delta_\mu h)\left(h^a{}_\nu\left[\partial^\nu h^\mu{}_a - \partial^\mu h^\nu{}_a\right]\right) + (\delta_\mu(\delta h))\left(h^a{}_\nu\left[\partial^\nu h^\mu{}_a - \partial^\mu h^\nu{}_a\right]\right)\right.$$

$$+\,(\delta_\mu h)\left(\delta h^a{}_\nu\left[\partial^\nu h^\mu{}_a - \partial^\mu h^\nu{}_a\right] + h^a{}_\nu\left[\partial^\nu\left(\delta h^\mu{}_a\right) - \partial^\mu\left(\delta h^\nu{}_a\right)\right]\right)$$

$$\left.+\,h\,\delta_\mu\left(\delta h^a{}_\nu\left[\partial^\nu h^\mu{}_a - \partial^\mu h^\nu{}_a\right] + h^a{}_\nu\left[\partial^\nu\left(\delta h^\mu{}_a\right) - \partial^\mu\left(\delta h^\nu{}_a\right)\right]\right)\right] + O\left(|\delta h|^2\right)$$

$$\tag{A11}$$

8. Superpotential $S_{ab}{}^\mu$:

$$S_{ab}{}^\mu + \delta S_{ab}{}^\mu = \frac{1}{2}\left(T_{ab}{}^\mu + \delta T_{ab}{}^\mu + T^\mu{}_{ba} + \delta T^\mu{}_{ba} - T^\mu{}_{ab} - \delta T^\mu{}_{ab}\right)$$

$$-\left(h^\mu{}_b + \delta h^\mu{}_b\right)\left(T_{\rho a}{}^\rho + \delta T_{\rho a}{}^\rho\right) + \left(h^\mu{}_a + \delta h^\mu{}_a\right)\left(T_{\rho b}{}^\rho + \delta T_{\rho b}{}^\rho\right)$$

$$\approx S_a{}^{\mu\nu} + \left[\frac{1}{2}\left(\delta T_{ab}{}^\mu + \delta T^\mu{}_{ba} - \delta T^\mu{}_{ab}\right) - \delta h^\mu{}_b T_{\rho a}{}^\rho - h^\mu{}_b \delta T_{\rho a}{}^\rho + \delta h^\mu{}_a T_{\rho b}{}^\rho\right.$$

$$\left.+\,h^\mu{}_a \delta T_{\rho b}{}^\rho\right] + O\left(|\delta h|^2\right)$$

$$\Rightarrow \delta S_{ab}{}^\mu \approx \left[\frac{1}{2}\left(\delta T_{ab}{}^\mu + 2\,\delta T^\mu{}_{ba}\right) - \delta h^\mu{}_b T_{\rho a}{}^\rho - h^\mu{}_b \delta T_{\rho a}{}^\rho + \delta h^\mu{}_a T_{\rho b}{}^\rho + h^\mu{}_a \delta T_{\rho b}{}^\rho\right] + O\left(|\delta h|^2\right)$$

$$\tag{A12}$$

In terms of $\delta h^a{}_\mu$, Equation (A12) becomes:

$$\delta S_{ab}{}^\mu \approx$$

$$\left[\frac{1}{2}\left(\partial_a\left(\delta h_b{}^\mu\right) - \partial_b\left(\delta h_a{}^\mu\right)\right) + \left(\partial_b\left(\delta h^\mu{}_a\right) - \partial_a\left(\delta h^\mu{}_b\right)\right) - \delta h^\mu{}_b\left(h^c{}_\rho\left[\partial_c h_a{}^\rho - \partial_a h_c{}^\rho\right]\right)\right.$$

$$-\,h^\mu{}_b\left(\delta h^c{}_\rho\left[\partial_c h_a{}^\rho - \partial_a h_c{}^\rho\right] + h^c{}_\rho\left[\partial_c\left(\delta h_a{}^\rho\right) - \partial_a\left(\delta h_c{}^\rho\right)\right]\right) + \delta h^\mu{}_a\left(h^c{}_\rho\left[\partial_c h_b{}^\rho - \partial_b h_c{}^\rho\right]\right)$$

$$\left.+\,h^\mu{}_a\left(\delta h^c{}_\rho\left[\partial_c h_b{}^\rho - \partial_b h_c{}^\rho\right] + h^c{}_\rho\left[\partial_c\left(\delta h_b{}^\rho\right) - \partial_b\left(\delta h_c{}^\rho\right)\right]\right)\right] + O\left(|\delta h|^2\right) \tag{A13}$$

9. Einstein tensor $\overset{\circ}{G}_{\mu\nu}$:

$$\mathring{G}_{ab} + \delta\mathring{G}_{ab} = \left(\mathring{G}_{\mu\nu} + \delta\mathring{G}_{\mu\nu}\right)\left(h_a{}^\mu + \delta h_a{}^\mu\right)\left(h_b{}^\nu + \delta h_b{}^\nu\right)$$

$$\approx \mathring{G}_{ab} + \left[\mathring{G}_{\mu\nu}\left(\delta h_a{}^\mu h_b{}^\nu + h_a{}^\mu \delta h_b{}^\nu\right) + \delta\mathring{G}_{\mu\nu}\left(h_a{}^\mu h_b{}^\nu\right)\right] + O\left(|\delta h|^2\right)$$

$$\Rightarrow \delta\mathring{G}_{ab} \approx \left[\mathring{G}_{\mu\nu}\left(\delta h_a{}^\mu h_b{}^\nu + h_a{}^\mu \delta h_b{}^\nu\right) + \delta\mathring{G}_{\mu\nu}\left(h_a{}^\mu h_b{}^\nu\right)\right] + O\left(|\delta h|^2\right). \tag{A14}$$

If $\mathring{G}_{\mu\nu} = \mathring{R}_{\mu\nu} - \frac{1}{2}g^{\sigma\rho}g_{\mu\nu}\mathring{R}_{\sigma\rho} = \mathring{R}_{\mu\nu} - \frac{\eta^{cd}\eta_{ab}}{2}\left[h_c{}^\sigma h_d{}^\rho h^a{}_\mu h^b{}_\nu\right]\mathring{R}_{\sigma\rho}$, then we obtain from Equation (A4):

$$\delta\mathring{G}_{\mu\nu} \approx \delta\mathring{R}_{\mu\nu} - \frac{\eta^{cd}\eta_{ab}}{2}\left[\left[h_c{}^\sigma h_d{}^\rho h^a{}_\mu h^b{}_\nu\right]\delta\mathring{R}_{\sigma\rho} + \left[\delta h_c{}^\sigma h_d{}^\rho h^a{}_\mu h^b{}_\nu + h_c{}^\sigma \delta h_d{}^\rho h^a{}_\mu h^b{}_\nu\right.\right.$$

$$\left.\left. + h_c{}^\sigma h_d{}^\rho \delta h^a{}_\mu h^b{}_\nu + h_c{}^\sigma h_d{}^\rho h^a{}_\mu \delta h^b{}_\nu\right]\mathring{R}_{\sigma\rho}\right] + O\left(|\delta h|^2\right) \tag{A15}$$

By substituting Equation (A15) into Equation (A14), we obtain that:

$$\delta\mathring{G}_{ab} \approx \left[\mathring{R}_{\mu\nu} - \frac{\eta^{cd}\eta_{ef}}{2}\left[h_c{}^\sigma h_d{}^\rho h^e{}_\mu h^f{}_\nu\right]\mathring{R}_{\sigma\rho}\right]\left(\delta h_a{}^\mu h_b{}^\nu + h_a{}^\mu \delta h_b{}^\nu\right)$$

$$+ \left(h_a{}^\mu h_b{}^\nu\right)\left[\delta\mathring{R}_{\mu\nu} - \frac{\eta^{cd}\eta_{ef}}{2}\left[\left[h_c{}^\sigma h_d{}^\rho h^e{}_\mu h^f{}_\nu\right]\delta\mathring{R}_{\sigma\rho}\right.\right.$$

$$\left.\left. + \left[\delta h_c{}^\sigma h_d{}^\rho h^e{}_\mu h^f{}_\nu + h_c{}^\sigma \delta h_d{}^\rho h^e{}_\mu h^f{}_\nu + h_c{}^\sigma h_d{}^\rho \delta h^e{}_\mu h^f{}_\nu + h_c{}^\sigma h_d{}^\rho h^e{}_\mu \delta h^f{}_\nu\right]\mathring{R}_{\sigma\rho}\right]\right]$$

$$+ O\left(|\delta h|^2\right) \tag{A16}$$

Now, if we have that $\mathring{R}_{\mu\nu} = h_k{}^\alpha h^m{}_\mu \mathring{R}^k{}_{m\alpha\nu}$, then Equation (A16) becomes

$$\delta\mathring{G}_{ab} \approx \left[h_k{}^\alpha h^m{}_\mu \mathring{R}^k{}_{m\alpha\nu} - \frac{\eta^{cd}\eta_{ef}}{2}\left[h_c{}^\sigma h_d{}^\rho h^e{}_\mu h^f{}_\nu\right]h_k{}^\alpha h^m{}_\sigma \mathring{R}^k{}_{m\alpha\rho}\right]\left(\delta h_a{}^\mu h_b{}^\nu + h_a{}^\mu \delta h_b{}^\nu\right)$$

$$+ \left(h_a{}^\mu h_b{}^\nu\right)\left[\left[\left(\delta h_k{}^\alpha h^m{}_\mu + h_k{}^\alpha \delta h^m{}_\mu\right)\mathring{R}^k{}_{m\alpha\nu} + h_k{}^\alpha h^m{}_\mu \delta\mathring{R}^k{}_{m\alpha\nu}\right]\right.$$

$$- \frac{\eta^{cd}\eta_{ef}}{2}\left[\left[h_c{}^\sigma h_d{}^\rho h^e{}_\mu h^f{}_\nu\right]\left[\left(\delta h_k{}^\alpha h^m{}_\sigma + h_k{}^\alpha \delta h^m{}_\mu\right)\mathring{R}^k{}_{m\alpha\rho} + h_k{}^\alpha h^m{}_\sigma \delta\mathring{R}^k{}_{m\alpha\rho}\right]\right.$$

$$+ \left[\delta h_c{}^\sigma h_d{}^\rho h^e{}_\mu h^f{}_\nu + h_c{}^\sigma \delta h_d{}^\rho h^e{}_\mu h^f{}_\nu + h_c{}^\sigma h_d{}^\rho \delta h^e{}_\mu h^f{}_\nu + h_c{}^\sigma h_d{}^\rho h^e{}_\mu \delta h^f{}_\nu\right]$$

$$\left.\left. \times h_k{}^\alpha h^m{}_\sigma \mathring{R}^k{}_{m\alpha\rho}\right]\right] + O\left(|\delta h|^2\right) \tag{A17}$$

For pure Minkowski spacetime, we have that $\mathring{R}^k{}_{m\alpha\rho} = 0$ by default and Equation (A17) reduces as:

$$\delta\mathring{G}_{ab} \approx \left(h_a{}^\mu h_b{}^\nu\right)\left[h_k{}^\alpha h^m{}_\mu \delta\mathring{R}^k{}_{m\alpha\nu} - \frac{\eta^{cd}\eta_{ef}}{2}\left[h_c{}^\sigma h_d{}^\rho h^e{}_\mu h^f{}_\nu\right]h_k{}^\alpha h^m{}_\sigma \delta\mathring{R}^k{}_{m\alpha\rho}\right] + O\left(|\delta h|^2\right). \tag{A18}$$

Equation (A18) is useful for Equations (40) and (48) and the energy-momentum stability test.

**Appendix B. General Perturbed Torsion-Based Field Equation via Linearization**

Here, we can also obtain the perturbed field equation (Equations (33) and (34)) using Equation (A9), with a matter contribution as follows:

$$\delta\mathcal{L} \quad \approx \quad \frac{1}{2\kappa}[\delta h\, f(T) + h\, f_T(T)\, \delta T] + \delta\mathcal{L}_{Matter} + O\left(|\delta h|^2\right) \tag{A19}$$

As for the non-perturbed FEs, we have here that $\delta\Theta_{(ab)} = \delta T_{ab} \equiv \frac{1}{2}\frac{\delta(\delta L_{Matt})}{\delta g_{ab}}$.

For the term $\frac{1}{2\kappa}\delta h\, f(T)$, we obtain by analogy with Equation (15) the following part (here, $\delta g_{ab} = 0$ for the orthonormal framework):

$$\begin{aligned}
\frac{\delta h\, f(T)}{2\kappa} \quad &\rightarrow \quad f_{TT}\left[\delta S_{(ab)}{}^{\mu}\, \partial_\mu T + S_{(ab)}{}^{\mu}\, \partial_\mu(\delta T)\right] + f_T\, \delta\overset{\circ}{G}_{ab} - \frac{g_{ab}}{2}\, f_T\, \delta T \quad & \text{Symmetric} \\
&\rightarrow \quad f_{TT}\left[S_{[ab]}{}^{\mu}\partial_\mu(\delta T) + \delta S_{[ab]}{}^{\mu}\partial_\mu T\right] & \text{Antisymmetric}
\end{aligned} \tag{A20}$$

At Equation (A20), we only perturb the physical quantities linked by $\delta h$, giving $\delta T$, $\delta\overset{\circ}{G}_{ab}$, and $\delta S_{ab}{}^{\mu}$. We do not perturb $f(T)$ and its derivatives.

For the term $\frac{1}{2\kappa}h\, f_T(T)\, \delta T$, we still obtain by analogy with Equation (15) the part (here again, $\delta g_{ab} = 0$):

$$\begin{aligned}
\frac{h\, f_T(T)\, \delta T}{2\kappa} \quad &\rightarrow \quad \left[f_{TTT}\, S_{(ab)}{}^{\mu}\partial_\mu T + f_{TT}\, \overset{\circ}{G}_{ab} + \frac{g_{ab}}{2}(f_T - T\, f_{TT})\right]\delta T \quad & \text{Symmetric} \\
&\rightarrow \quad f_{TTT}\left[S_{[ab]}{}^{\mu}\partial_\mu T\right]\delta T & \text{Antisymmetric}
\end{aligned} \tag{A21}$$

At Equation (A21), we only change $f(T) \rightarrow f_T(T)\, \delta T$, $f_T(T) \rightarrow f_{TT}(T)\, \delta T$, and $f_{TT}(T) \rightarrow f_{TTT}(T)\, \delta T$. We does not perturb the physical quantities themselves.

By adding the Equations (A20) and (A21), we obtain exactly at the first order the Equations (33) and (34). This is the sign that the linearization of gravity and the direct perturbation of the field equation described by Equation (15) are both equivalent. Through these two methods, we obtain the field equation described by Equations (33) and (34), which is in the order of things.

**Appendix C. The Derivation of Minkowski Spacetime Symmetries: Conditions for Stability**

In order to shorten the text, we put in this appendix some long calculations that are necessary for the results of Sections 5.1 and 5.2.

*Appendix C.1. Rotation/Boost Perturbation*

1.  Torsion scalar perturbation $\delta T$: by using Equation (A8) and by substituting Equation (39) inside, we obtain the expression:

$$
\begin{aligned}
\delta T =& \frac{1}{4}\Bigg[\left(\partial_\mu\left(\lambda^a{}_b\,h^b{}_\nu\right)-\partial_\nu\left(\lambda^a{}_b\,h^b{}_\mu\right)\right)\left(\partial^\mu h_a{}^\nu-\partial^\nu h_a{}^\mu\right)+\left(\partial_\mu h^a{}_\nu-\partial_\nu h^a{}_\mu\right)\\
&\times\left(\partial^\mu\left(\lambda_a{}^b\,h_b{}^\nu\right)-\partial^\nu\left(\lambda_a{}^b\,h_b{}^\mu\right)\right)\Bigg]+\frac{1}{2}\Bigg[\left(\partial_\mu\left(\lambda^a{}_b\,h^b{}_\nu\right)-\partial_\nu\left(\lambda^a{}_b\,h^b{}_\mu\right)\right)\left(\partial^\nu h^\mu{}_a-\partial^\mu h^\nu{}_a\right)\\
&+\left(\partial_\mu h^a{}_\nu-\partial_\nu h^a{}_\mu\right)\left(\partial^\nu\left(\lambda^b{}_a\,h^\mu{}_b\right)-\partial^\mu\left(\lambda^b{}_a\,h^\nu{}_b\right)\right)\Bigg]\\
&-\Bigg[\left(\lambda_a{}^b\,h_b{}^\nu\left[\partial_\mu h^a{}_\nu-\partial_\nu h^a{}_\mu\right]+h_a{}^\nu\left[\partial_\mu\left(\lambda^a{}_b\,h^b{}_\nu\right)-\partial_\nu\left(\lambda^a{}_b\,h^b{}_\mu\right)\right]\right)\left(h^a{}_\rho\left(\partial^\rho h^\mu{}_a-\partial^\mu h^\rho{}_a\right)\right)\\
&+\left(h_a{}^\nu\left[\partial_\mu h^a{}_\nu-\partial_\nu h^a{}_\mu\right]\right)\left(\lambda^a{}_b h^b{}_\rho\left(\partial^\rho h^\mu{}_a-\partial^\mu h^\rho{}_a\right)+h^a{}_\rho\left(\partial^\rho\left(\lambda^b{}_a\,h^\mu{}_b\right)-\partial^\mu\left(\lambda^b{}_a\,h^\rho{}_b\right)\right)\right)\Bigg]\\
&+O\left(|\delta h|^2\right)\\
\rightarrow&\,0
\end{aligned}
$$

(A22)

We need to impose $T^a{}_{\mu\nu}=\partial_\mu h^a{}_\nu-\partial_\nu h^a{}_\mu\rightarrow 0$ to obtain the final result for Equation (A22).

2.  Superpotential perturbation $\delta S_{ab}{}^\mu$: by using Equation (A13) and by substituting Equation (39) inside, we obtain the expression:

$$
\begin{aligned}
\delta S_{ab}{}^\mu&\\
=&\Bigg[\frac{1}{2}\left(\partial_a\left(\lambda^c_b\,h_c{}^\mu\right)-\partial_b\left(\lambda_a{}^c\,h_c{}^\mu\right)\right)+\left(\partial_b\left(\lambda^c_a\,h^\mu_c\right)-\partial_a\left(\lambda^c_b\,h^\mu_c\right)\right)-\lambda^e_b\,h^\mu_e\left(h^c_\rho\left[\partial_c h_a{}^\rho-\partial_a h_c{}^\rho\right]\right)\\
&-h^\mu_b\left(\lambda^c_e\,h^e_\rho\left[\partial_c h_a{}^\rho-\partial_a h_c{}^\rho\right]+h^c_\rho\left[\partial_c\left(\lambda_a^f h^\rho_f\right)-\partial_a\left(\lambda_c^f h^\rho_f\right)\right]\right)\\
&+\lambda^e_a\,h^\mu_e\left(h^c_\rho\left[\partial_c h_b{}^\rho-\partial_b h_c{}^\rho\right]\right)+h^\mu_a\lambda^c_e h^e_\rho\left[\partial_c h_b{}^\rho-\partial_b h_c{}^\rho\right]\\
&+h^\mu_a\,h^c_\rho\left[\partial_c\left(\lambda^f_b h_f{}^\rho\right)-\partial_b\left(\lambda^f_c h_f{}^\rho\right)\right]\Bigg]+O\left(|\delta h|^2\right)\\
\rightarrow&\Bigg[\frac{1}{2}\left(\partial_a\left(\lambda^c_b\,h_c{}^\mu\right)-\partial_b\left(\lambda_a{}^c\,h_c{}^\mu\right)\right)+\left(\partial_b\left(\lambda^c_a\,h^\mu_c\right)-\partial_a\left(\lambda^c_b\,h^\mu_c\right)\right)\\
&-h^\mu_b\,h^c_\rho\left[\partial_c\left(\lambda_a^f h_f{}^\rho\right)-\partial_a\left(\lambda_c^f h_f{}^\rho\right)\right]+h^\mu_a\,h^c_\rho\left[\partial_c\left(\lambda^f_b h_f{}^\rho\right)-\partial_b\left(\lambda^f_c h_f{}^\rho\right)\right]\Bigg]\\
&+O\left(|\delta h|^2\right)\qquad\text{by applying Equation (42) (the zero torsion criteria).}\\
\rightarrow&\,0.
\end{aligned}
$$

(A23)

We need to impose $\delta T^a{}_{\mu\nu}=\partial_\mu\left(\lambda^a{}_c\,h^c{}_\nu\right)-\partial_\nu\left(\lambda^a{}_c\,h^c{}_\mu\right)\rightarrow 0$ to obtain the final result for Equation (A23).

3.  Weitzenbock connection perturbation $\delta\Gamma^\rho{}_{\nu\mu}$: from the null covariant derivative criteria, we make the following derivation as:

$$
\begin{aligned}
0=\nabla_\mu\,\delta h^a{}_\nu\;=&\;\nabla_\mu\left(\lambda^a{}_b\,h^b{}_\nu\right)=\partial_\mu\,\delta h^a{}_\nu-\Gamma^\rho{}_{\nu\mu}\,\delta h^a{}_\rho-\delta\Gamma^\rho{}_{\nu\mu}h^a{}_\rho\\
=&\;\partial_\mu\left(\lambda^a{}_b\,h^b{}_\nu\right)-\left(h_c{}^\rho\,\partial_\mu\,h^c{}_\nu\right)\left(\lambda^a{}_b\,h^b{}_\rho\right)-\delta\Gamma^\rho{}_{\nu\mu}h^a{}_\rho\\
\Rightarrow&\;\delta\Gamma^\rho{}_{\nu\mu}=h_a{}^\rho\left[\partial_\mu\left(\lambda^a{}_b\,h^b{}_\nu\right)-\left(h_c{}^\sigma\,\partial_\mu\,h^c{}_\nu\right)\left(\lambda^a{}_b\,h^b{}_\sigma\right)\right],
\end{aligned}
$$

(A24)

where $\Gamma^\rho{}_{\nu\mu} = h_c{}^\rho \, \partial_\mu h^c{}_\nu$ is the Weitzenbock connection for a proper frame.

*Appendix C.2. General Linear Perturbation*

1. The torsion scalar perturbation $\delta T$:

$$
\begin{aligned}
\delta T = &\frac{1}{4}\left[\left(\partial_\mu\left(\lambda^a{}_b\,h^b{}_\nu + \epsilon^a{}_\nu\right) - \partial_\nu\left(\lambda^a{}_b\,h^b{}_\mu + \epsilon^a{}_\mu\right)\right)\left(\partial^\mu h_a{}^\nu - \partial^\nu h_a{}^\mu\right) + \left(\partial_\mu h^a{}_\nu - \partial_\nu h^a{}_\mu\right)\right. \\
&\left. \times \left(\partial^\mu\left(\lambda_a{}^b\,h_b{}^\nu + \epsilon_a{}^\nu\right) - \partial^\nu\left(\lambda_a{}^b\,h_b{}^\mu + \epsilon_a{}^\mu\right)\right)\right] \\
&+\frac{1}{2}\left[\left(\partial_\mu\left(\lambda^a{}_b\,h^b{}_\nu + \epsilon^a{}_\nu\right) - \partial_\nu\left(\lambda^a{}_b\,h^b{}_\mu + \epsilon^a{}_\mu\right)\right)\left(\partial^\nu h^\mu{}_a - \partial^\mu h^\nu{}_a\right)\right. \\
&\left. + \left(\partial_\mu h^a{}_\nu - \partial_\nu h^a{}_\mu\right)\left(\partial^\nu\left(\lambda^b{}_a\,h^\mu{}_b + \epsilon^\mu{}_a\right) - \partial^\mu\left(\lambda^b{}_a\,h^\nu{}_b + \epsilon^\nu{}_a\right)\right)\right] \\
&-\left[\left(\left(\lambda_a{}^b\,h_b{}^\nu + \epsilon_a{}^\nu\right)\left[\partial_\mu h^a{}_\nu - \partial_\nu h^a{}_\mu\right] + h_a{}^\nu\left[\partial_\mu\left(\lambda^a{}_b\,h^b{}_\nu + \epsilon^a{}_\nu\right) - \partial_\nu\left(\lambda^a{}_b\,h^b{}_\mu + \epsilon^a{}_\mu\right)\right]\right)\right. \\
&\times \left(h^a{}_\rho\left(\partial^\rho h^\mu{}_a - \partial^\mu h^\rho{}_a\right)\right) + \left(h_a{}^\nu\left[\partial_\mu h^a{}_\nu - \partial_\nu h^a{}_\mu\right]\right) \\
&\left. \times \left(\left(\lambda^a{}_b h^b{}_\rho + \epsilon^a{}_\rho\right)\left(\partial^\rho h^\mu{}_a - \partial^\mu h^\rho{}_a\right) + h^a{}_\rho\left(\partial^\rho\left(\lambda^b{}_a\,h^\mu{}_b + \epsilon^\mu{}_a\right) - \partial^\mu\left(\lambda^b{}_a\,h^\rho{}_b + \epsilon^\rho{}_a\right)\right)\right)\right] \\
&+O\left(|\delta h|^2\right) \\
&\to 0.
\end{aligned}
\tag{A25}
$$

We again need to impose $T^a{}_{\mu\nu} = \partial_\mu h^a{}_\nu - \partial_\nu h^a{}_\mu \to 0$ as for Equation (A22) to obtain Equation (A25).

2. The superpotential perturbation $\delta S_{ab}{}^\mu$ is expressed as:

$$
\begin{aligned}
\delta S_{ab}{}^\mu = &\left[\frac{1}{2}\left(\partial_a\left(\lambda^c_b\,h_c{}^\mu + \epsilon^\mu{}_b\right) - \partial_b\left(\lambda^c_a\,h_c{}^\mu + \epsilon^\mu{}_a\right)\right) + \left(\partial_b\left(\lambda^c_a\,h^\mu{}_c + \epsilon^\mu{}_a\right) - \partial_a\left(\lambda^c_b\,h^\mu{}_c + \epsilon^\mu{}_b\right)\right)\right. \\
&-\left(\lambda^e_b\,h^\mu{}_e + \epsilon^\mu{}_b\right)\left(h^c{}_\rho\left[\partial_c h_a{}^\rho - \partial_a h_c{}^\rho\right]\right) \\
&-h^\mu{}_b\left(\left(\lambda^c_e\,h^e{}_\rho + \epsilon^c{}_\rho\right)\left[\partial_c h_a{}^\rho - \partial_a h_c{}^\rho\right] + h^c{}_\rho\left[\partial_c\left(\lambda^f_a\,h^\rho{}_f + \epsilon^\rho{}_a\right) - \partial_a\left(\lambda^f_c\,h^\rho{}_f + \epsilon^\rho{}_c\right)\right]\right) \\
&+\left(\lambda^e_a\,h^\mu{}_e + \epsilon^\mu{}_a\right)\left(h^c{}_\rho\left[\partial_c h_b{}^\rho - \partial_b h_c{}^\rho\right]\right) + h^\mu{}_a\left(\lambda^c_e\,h^e{}_\rho + \epsilon^c{}_\rho\right)\left[\partial_c h_b{}^\rho - \partial_b h_c{}^\rho\right] \\
&\left. + h^\mu{}_a\,h^c{}_\rho\left[\partial_c\left(\lambda^f_b\,h^\rho{}_f + \epsilon^\rho{}_b\right) - \partial_b\left(\lambda^f_c\,h^\rho{}_f + \epsilon^\rho{}_c\right)\right]\right] + O\left(|\delta h|^2\right) \\
\to &\left[\frac{1}{2}\left(\partial_a\left(\lambda^c_b\,h_c{}^\mu + \epsilon^\mu{}_b\right) - \partial_b\left(\lambda^c_a\,h_c{}^\mu + \epsilon^\mu{}_a\right)\right) + \left(\partial_b\left(\lambda^c_a\,h^\mu{}_c + \epsilon^\mu{}_a\right) - \partial_a\left(\lambda^c_b\,h^\mu{}_c + \epsilon^\mu{}_b\right)\right)\right. \\
&-h^\mu{}_b\,h^c{}_\rho\left[\partial_c\left(\lambda^f_a\,h^\rho{}_f + \epsilon^\rho{}_a\right) - \partial_a\left(\lambda^f_c\,h^\rho{}_f + \epsilon^\rho{}_c\right)\right] \\
&\left. + h^\mu{}_a\,h^c{}_\rho\left[\partial_c\left(\lambda^f_b\,h^\rho{}_f + \epsilon^\rho{}_b\right) - \partial_b\left(\lambda^f_c\,h^\rho{}_f + \epsilon^\rho{}_c\right)\right]\right] + O\left(|\delta h|^2\right) \\
\to &\,0,
\end{aligned}
\tag{A26}
$$

where we apply Equation (42) and we respect the $\partial_a\epsilon_b{}^\mu = \partial_b\epsilon_a{}^\mu = 0$ condition.

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
