# Peer review of "Teleparallel Minkowski Spacetime with Perturbative Approach for Teleparallel Gravity on a Proper Frame"

_universe, doi:10.3390/universe9050232_

Round 1
Reviewer 1 Report
This paper aims to develop the perturbation theory in teleparallel gravity on the fundamental quantities (fields) describing the theory. This topic has been modestly researched by a small part of the teleparallel community, and it has notable importance regarding physical predictions and the theoretical consistency of the theories. My overall impression of this manuscript is that the research presented here is scientifically sound and valid and deserves publication. However the authors are missing an important part of the recent research in the field, and connecting their results with others in the literature. Some issues in this direction, and other important matters that I would like to point out, are:
1. The authors establish some definitions of what is a Minkowski spacetime (subsection 3.1.1). In particular, the requirements of null torsion and nonmetricity tensors, on top of the vanishing curvature, have been first used for such definition in Beltran-Koivisto 2104.05566, section 6. Since the authors have not cited this paper in 3.1.1, I believe it is pertinent that they revise the arguments and computations there. This is also concerning the criticism of the authors to Ref [22] in the introduction, since in that paper it is taken a boosted coframe with nonvanishing torsion tensor that it is still a Minkowski solution of f(T) equations of motion, conflicting with the definition of Minkowski geometry made in 2104.05566 and in their paper. By the way, the authors of Ref [23] have been partially wrongly addressed in the fourth chapter of their paper, and they should correspond to Beltran-Golovnev-Koivisto-Veermae instead of Golovnev-Guzman.
2. It is surprising that the authors do not cite, or are not aware, of fundamental and well cited references regarding the choice of the spin connection Eq.(8), as for instance 1701.06271, where the case for f(T) has also been made. They should refer to all the previous works where it has been established.
3. In subsection 2.5 Eq.(9) it is presented "the" torsion scalar, which is called "the" one because of certain properties, for instance, being equivalent to the Ricci scalar up to a boundary term, therefore being the only combination of the c_i parameters (as in Eq.(10)) that give an equivalent to general relativity. Also, in Eq.(12) it should be mentioned with respect to what field it is considered the principle of least-action. The variation should be taken with respect to the frame or coframe components, but the indices of the equations are purely tangent-space ones, therefore confusing a careless reader.
4. It's not clear for me what is exactly the point in the results presented in 4.3 and 4.4, other than obtaining the perturbations at first order with a background torsion scalar as constant/zero. It would be more useful to study the properties of vanishing the perturbed torsion scalar to zero, not setting it zero at the background level. This is relevant in the discussion concerning the nonperturbative behavior of f(T) gravity in the branching of solutions. See for instance the results presented in 1105.5934, and the branching in the algorithm more clearly identified in 1802.02130 and 2006.15303. An interesting discussion that could derive from the results of the authors, if they considered carefully the conclusions in the papers mentioned, is that solutions with constant torsion scalar (or even time dependent one) contain both Minkowski and FLRW spacetime. However, a point that is not clear is if a constant torsion scalar continues being constant at first order in perturbations (or any higher order). Unfortunately, this is not possible to derive from their analysis, since they have set the torsion scalar to zero at the background level already.
5. The framework for generic perturbations in the tetrad formalism has been developed for cosmological perturbations in 1808.05565, as an improvement to previous research where it wasn't considered the most general perturbations (with infinitesimal boosts and rotations). Since perturbations to Minkowski can be easily obtained from cosmological perturbation equations by setting a(t)=1, it is unclear why the authors haven't mentioned this and previous work, and made some possible comparison. Most certainly, the parametrization of perturbations with the parameters \lambda^{a}_{b} is in one-to-one correspondence with the parametrization of perturbations made in 1808.05565.
6. Finally, I haven't seen any comments regarding the strong coupling issues of f(T) gravity, and how their perturbation analysis could be trusted when there isn't a clear answer regarding the number of degrees of freedom. From their results I could expect that some factors multiplying kinetic terms for perturbations vanish under certain points of the phase space. Since they take the most trivial tetrad, I could expect that the extra degrees of freedom do not show at the perturbative level in their analysis. The conditions of stability derived seem to be equivalent to the most trivial Minkowski background, that is, e^{a}_{\mu}=(-1,1,1,1), which certainly vanish not only the torsion tensor but the superpotential and the torsion scalar. Therefore it is not clear from their analysis if their perturbations apply for f(T) or only for the case of f(T) when T=const or 0. If the latter, then the perturbations describe just GR/TEGR, and no new information or insights could be extract from their analysis.
I will consider acceptance of this paper after the authors make a major revision of their manuscript in the lines of the comments written above.
Reviewer 2 Report
After reading this work several times and partially carrying out in some detail the results presented by the authors, I must say that the article is well written and organized. It presents an extensive bibliography and also establishes from the beginning the main objective of the study and the conventions in the notation used. The first few sections are illuminating, as any reader has at their disposal a concise but excellent introduction to Minkowski geometry in teleparallel gravity as well as to the theory itself.
The remaining sections present a very complete, detailed and meticulous analysis of the perturbation scheme applied in the framework of f(T) teleparallel theory. This work represents a real advance in the direction of perturbative studies in modified gravity theories and therefore should be published imminently in its current form.
Regards, the referee.
Reviewer 3 Report
The authors study perturbative approach around Minkowski spacetime within Teleparallel Gravity and obtain perturbed equation of motion followed by perturbations of geometric objects like like curvature, energy-momentum tensor or torsions tensor. Then they use obtained results for analysing stability of Minkowski space-time.
The manuscript is generally well written, in a clear way, the results are presented in a logical way. However, this clear way in a very hermetic way, or more like a mathematical paper not physical one. There are very lengthy formulae that does not add much in order to understand the results. Introduction is missing general description of Teleparallel Gravity, in the conclusions they write what they want to study later, not what can be exactly studied and how relevant it is to the current analysis. I don't find all that convincing.
I don't find much physical content and I think the content should be rather treated as formal derivation than research in physics.
I suggest that authors submit that paper to a more mathematical journal, in the area of mathematical physics, I think it fits well the type of research presented there.
After that, the physical results promised by authors in conclusions, they may apply their formalism to more concrete problems. Although they don't write, which "concrete" problems, only vague statements like "more concrete problems of the astrophysical and/or cosmological type" or that "There are some works which are coming much later about these subjects."
I think application of their analysis described in the current manuscript to the cosmological problems might be appropriate to be published in a physical journal, like Universe, but the current very mathematical manuscript not.
Of course the authors could rewrite the whole manuscript trying to expose physics underneath, but is it worth their time? I leave the answer to the editor and authors themselves. My opinion is that it is not valid for publication in it current form, due to reasons described above.
Round 2
Reviewer 1 Report
The changes made by the authors, new comparisons and citations are sufficient in order to guarantee publication in Universe. However, some extra comments on the newly added paragraphs:
1. Several typos in the new paragraph in the introduction, such as "but but", "the any", "theory perturbations".
2. What do the authors mean by the phrase "condition and singularity" added at the end of page 6?
3. I also have a comment on the penultimate paragraph in the new version of the paper. Actually the story is more complicated and in ref.46 it is claimed that the Dirac algorithm has several branches, and in one of them the dof is 2, as in GR. Other pathological cases have 4 and 0 dof in total. And the most general case seems to have 5 dof (3 extra dof). In ref.45 only the branch with 5dof has been properly identified. In ref.47 it has been claimed that the branch where the dof was claimed to be 2 in the previous paper, it is actually 3 (1 extra dof). Due to a mistake in ref.47, the branch with 5 dof has not been properly identified. In ref.22 there is evidence supporting 1 extra dof in f(T) gravity at the perturbative level, but more dof could be hidden at higher-order perturbations. A summary on the discussion of the outcomes of the different Hamiltonian formalisms can be found in 2012.14408, in case the authors would like to be more accurate in their review of the unresolved challenges.
Reviewer 3 Report
I have read an improved versioning found it better than the original one, thanks to detailed remarks of the other referee. With the added descriptions, both in the introduction, in deriving field equations and perturbed field equations, the text became less hermetic and both easier and more interesting to read. The widely expanded chapter on perturbations on trivial coframes now add a lot to the context of the paper.
As it is clear now, that the paper is submitted to the special issue on mathematical cosmology, I withdraw my previous criticism about too mathematical character. This information was not provided during a first round.
However, I still find several formulae (like eq. 41, 43, 45), too long to merit being inserted in the main text and in my opinions long derivations should be put in the appendix. Anyway, the authors consider limits of these equations and the formulae themselves are not of the crucial importance. They exhibit the whole procedure and might be presented, but only as additional material possibly in the appendix.
After clearing a bit main text from long formulae, by putting them in the appendix for example, the manuscript could be publishable in the Universe in my opinion.
